# Proteogenomics and Hi-C reveal transcriptional dysregulation in high hyperdiploid childhood acute lymphoblastic leukemia

Minjun Yang [1], Mattias Vesterlund [2], Ioannis Siavelis[2], Larissa H. Moura-Castro[1], Anders Castor[3], Thoas Fioretos[1], Rozbeh Jafari [2], Henrik Lilljebjörn [1], Duncan T. Odom [4,5], Linda Olsson[1,6], Naveen Ravi[1], Eleanor L. Woodward[1], Louise Harewood[4,7], Janne Lehtiö [2] & Kajsa Paulsson [1]

Hyperdiploidy, i.e. gain of whole chromosomes, is one of the most common genetic features of childhood acute lymphoblastic leukemia (ALL), but its pathogenetic impact is poorly understood. Here, we report a proteogenomic analysis on matched datasets from genomic profiling, RNA-sequencing, and mass spectrometry-based analysis of >8,000 genes and proteins as well as Hi-C of primary patient samples from hyperdiploid and *ETV6/RUNX1*-positive pediatric ALL. We show that CTCF and cohesin, which are master regulators of chromatin architecture, display low expression in hyperdiploid ALL. In line with this, a general genome-wide dysregulation of gene expression in relation to topologically associating domain (TAD) borders were seen in the hyperdiploid group. Furthermore, Hi-C of a limited number of hyperdiploid childhood ALL cases revealed that 2/4 cases displayed a clear loss of TAD boundary strength and 3/4 showed reduced insulation at TAD borders, with putative leukemogenic effects.

[1] Division of Clinical Genetics, Department of Laboratory Medicine, Lund University, SE-221 84 Lund, Sweden. [2] Department of Oncology-Pathology, Science for Life Laboratory and Karolinska Institute, Clinical Proteomics Mass Spectrometry, SE-171 21 Stockholm, Sweden. [3] Department of Pediatrics, Skåne University Hospital, Lund University, SE-221 85 Lund, Sweden. [4] Cancer Research UK Cambridge Institute (CRUK-CI), University of Cambridge, Li Ka Shing Centre, Cambridge CB2 0RE, UK. [5] German Cancer Research Center (DKFZ), Division of Signaling and Functional Genomics, 69120 Heidelberg, Germany. [6] Department of Clinical Genetics and Pathology, Office for Medical Services, Division of Laboratory Medicine, SE-221 85 Lund, Sweden. [7] Precision Medicine Centre of Excellence, Queen's University Belfast, 97 Lisburn Road, Belfast BT9 7AE, UK. These authors contributed equally: Minjun Yang, Mattias Vesterlund. Correspondence and requests for materials should be addressed to J.L. (email: janne.lehtio@ki.se) or to K.P. (email: kajsa.paulsson@med.lu.se)

Aneuploidy, i.e., changes in chromosome numbers, is one of the most common phenomena in cancer cells. In spite of the huge efforts that have gone into understanding the impact of somatic genetic events in cancer, the effects of aneuploidy in tumorigenesis remain poorly understood. In fact, it is even debated whether aneuploidy in itself may be a driver event or if it is a passenger event without consequences in tumor development[1].

High hyperdiploid (51–67 chromosomes) pediatric B-cell precursor acute lymphoblastic leukemia (BCP ALL) is one of the most common malignancies in early childhood and is associated with a median age at diagnosis of 3–5 years, a low white blood cell count, and a favorable prognosis on contemporary treatment protocols[2]. Genetically, its defining feature is a non-random aneuploidy consisting of extra chromosomes, most commonly X, 4, 6, 10, 14, 17, 18, and 21. Approximately half of cases also harbor mutations in the RTK–RAS pathway, primarily *KRAS*, and 20% have mutations in histone modifiers such as *CREBBP*, in addition to microdeletions of various genes involved in B-cell differentiation/cell cycle control[3,4]. However, these additional aberrations are seen only in a subset of the cases, are sometimes gained or lost at relapse, and, when occurring, are frequently subclonal, whereas the aneuploidy is uniformly present[3,4]. Furthermore, we and others have shown that the chromosomal gains in these cases are early and likely leukemia-initiating aberrations, often arising several years before overt disease[3,5]. Taken together, available data strongly indicate that the aneuploidy is the main driver event in this type of leukemia, but the underlying leukemogenic mechanism remains unclear.

Previous studies of the RNA expression pattern in high hyperdiploid ALL have revealed a general upregulation of genes on the gained chromosomes, hinting that dosage effects may occur[3,6,7]. However, no detailed analysis of how this may affect leukemogenesis has yet been published and it remains unknown how the chromosomal gains may cause the development of leukemia. Additionally, since genes are subject to post-transcriptional control, the RNA expression level of a gene may not be directly transferable to the protein level. To address this, we performed a proteogenomic analysis of a series of pediatric BCP-ALL, including high hyperdiploid and diploid/near-diploid *ETV6/RUNX1*-positive cases, aiming to determine the effects of aneuploidy. Besides demonstrating that the characteristic extra chromosomes have an impact on the transcriptome and proteome, we also present data suggesting that hyperdiploid leukemia cases harbor aberrant chromatin organization that causes genome-wide transcriptional dysregulation. Taken together, our data give insight into the leukemogenesis of this common and clinically important pediatric leukemia.

## Results

**Proteogenomic analysis of childhood BCP ALL.** This study comprised mass spectrometry (MS)-based analysis of the proteome, whole genome and/or whole exome sequencing (WGS/WES) for somatic mutations and structural events, SNP array analysis for copy number assessment, and RNA-sequencing (RNA-seq) for RNA expression in childhood BCP ALL. In total, 48 high hyperdiploid and 41 *ETV6/RUNX1*-positive cases were investigated; the cohort analyzed with MS comprised eighteen high hyperdiploid and nine *ETV6/RUNX1*-positive pediatric ALL cases, as ascertained by chromosome banding, fluorescence in situ hybridization (FISH), SNP array analysis and reverse transcriptase-PCR for the fusion transcript (Fig. 1a and Supplementary Data 1 and 2).

MS data were generated via high-resolution isoelectric focusing liquid chromatography mass spectrometry (HiRIEF LC-MS/MS)

workflow together with isobaric labeling (TMT10) for relative quantification between tumors[8]. In total, 10,981 proteins originating from 10,138 genes were identified at 1% protein false discovery rate (FDR) based on 174,966 unique peptides (Fig. 1b, Supplementary Table 1 and Supplementary Data 3). For all quantitative proteome analyses, we used a gene symbol centric subset of 8480 genes that were quantified in each of the 27 tumors (Fig. 1b).

Genomic and transcriptomic variation was observed at the peptide level by searching HiRIEF LC-MS/MS spectra against a customized sequence database, which included both human RefSeq protein as well as somatic mutant and fusion sequences derived from WGS/WES and RNA-seq. Although many SNPs were seen in the protein dataset, none of the somatic mutations could be detected. The *ETV6/RUNX1* fusion could be identified at the protein level in all nine cases from this subgroup.

**Proteome analyses give improved biological insight in cancer.** For 8222 (97%) of the 8480 proteins detected by HiRIEF LC-MS/MS, the expression of the corresponding mRNA could be ascertained by rRNA-depleted RNA-seq (RiboZero RNA-seq) of the same samples (Supplementary Data 4 and 5). Expression levels were positively correlated for most (75%) mRNA–protein pairs across the 27 samples, with 22% showing significant correlation (multiple-test adjusted $P \le 0.05$) and a mean Spearman's correlation coefficient of 0.24 (Fig. 1c). This is similar to what has previously been reported in colorectal cancer[9], but lower than in ovarian cancer and breast cancer[10,11]. When the correlation scores were ascertained for different KEGG pathways, scores were highest for specialized pathways, such as hematopoietic cell lineage and amino acid metabolism, and lowest for house-keeping functions, e.g., ribosomal and spliceosomal processes (Fig. 1d). This is in line with the previous studies[9–11] and demonstrates that the level of expression of mRNA is not always directly translatable to the protein level. To further test the post-translational gene regulation effect on leukemia samples, we performed pairwise correlation of gene/protein abundance for all 8222 proteins. Similar to results previously obtained from the TCGA and CPTAC datasets[12], the correlation score for pairs of proteins involved in the same protein complex displayed a degree of co-regulation (mean Spearman's correlation coefficient = 0.19) that was significantly higher than that observed for random pairs (mean Spearman's correlation coefficient = 0) (Supplementary Fig. 1). A similar co-regulation effect could be seen at the transcript level (mean Spearman's correlation coefficient = 0.16), but the correlation was significantly lower than at the protein level (two-tailed Mann–Whitney $U$-test $P = 3.80e{-}20$).

To explore the details of this relatively low correlation between mRNA expression and protein abundance, we investigated several aspects of mRNA and protein regulation. A global comparison of stable and unstable mRNAs and their corresponding proteins[13] revealed significantly higher correlation for genes with similar stability on both the mRNA and protein levels (Supplementary Fig. 2). Next, we investigated the potential impact of miRNA-targeting. Genes regulated by miRNAs[14] displayed significantly lower mRNA and protein correlations, showing a role for post-transcriptional RNA regulation (Supplementary Fig. 2). We also observed that protein level regulation by the ubiquitin–proteasome pathway[15] affected the correlations since genes with low mRNA–protein correlations were significantly more frequently targeted by the proteasome (Supplementary Fig. 2). Consistent with this result, an analysis of the protein degradation rate also showed that genes with low mRNA–protein correlations were enriched among rapidly degrading proteins[16] (Supplementary Fig. 2). Interestingly, we observed that protein

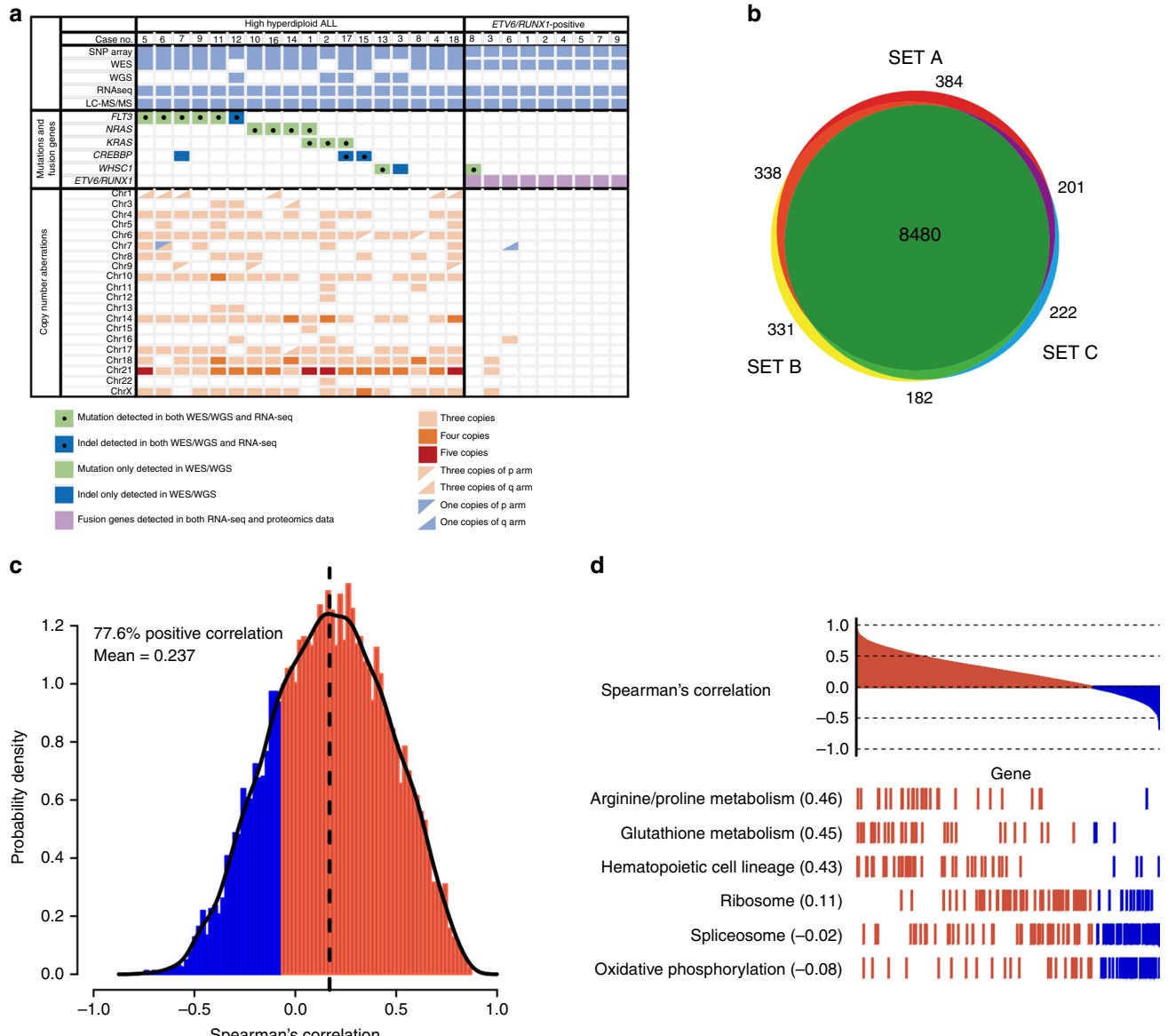

**Fig. 1** Proteogenomic study of childhood B-cell precursor acute lymphoblastic leukemia (ALL). **a** Genomic landscape of 27 childhood ALL included in the proteogenomic analysis. All cases were disomic for chromosomes 2, 19, and 20. **b** Numbers of proteins overlapping across the 3 TMT-sets. **c** Spearman's rank order correlation between mRNA and protein abundance. The correlation was positive for 77.6% mRNA-protein pairs in the whole cohort of cases with a mean Spearman's correlation coefficient of 0.24. Approximately 23% mRNA-protein pairs showed significant correlation (multiple-test adjusted $P \leq$ 0.05). **d** When investigating different biological processes, mRNA and protein levels displayed the highest correlation for specialized pathways, such as hematopoietic cell lineage and amino acid metabolism, and lowest for house-keeping functions, e.g., ribosomal and spliceosomal processes

subcellular localization had an effect as cytosolic and proteins residing in the plasma membrane, endoplasmic reticulum and the Golgi (i.e., secretory proteins) exhibited increased mRNA-protein correlations whilst nuclear and mitochondrial proteins did not (Supplementary Fig 2). Finally, we also observed that mRNA and proteins that were differentially expressed between hyperdiploid and *ETV6/RUNX1*-positive leukemia had higher mRNA-protein correlations (Supplementary Fig. 2). This fits with the observation in Orre et al.[17] that the secretory protein subset provides a better separation of cell lineages and cell types compared to nuclear proteins. Phenotypic genes thus seem to be more highly correlated on the mRNA–protein levels. Taken together, our analyses show that multiple factors contribute to lowering the correlation between mRNA and protein levels. Thus, proteome analyses are likely to give more biologically relevant data on

dysregulated pathways in cancer than RNA expression analyses alone.

**Impact of copy number events**. To study the impact of the extra chromosomes in high hyperdiploid ALL, we first compared the mean RNA and protein expression according to copy number in high hyperdiploid ALL. This clearly showed that the hyperdiploidy is associated with dosage effects, i.e., a generally increasing expression of genes and proteins with higher copy number (also termed *cis* effects; Fig. 2a). Notably, however, not all genes and proteins were affected in this way; approximately 16% (283/2,080) of genes and 25% (523/2,080) of proteins instead showed negative correlation with copy number. Thus, copy number gain does not always lead to increased expression, in particular at the protein

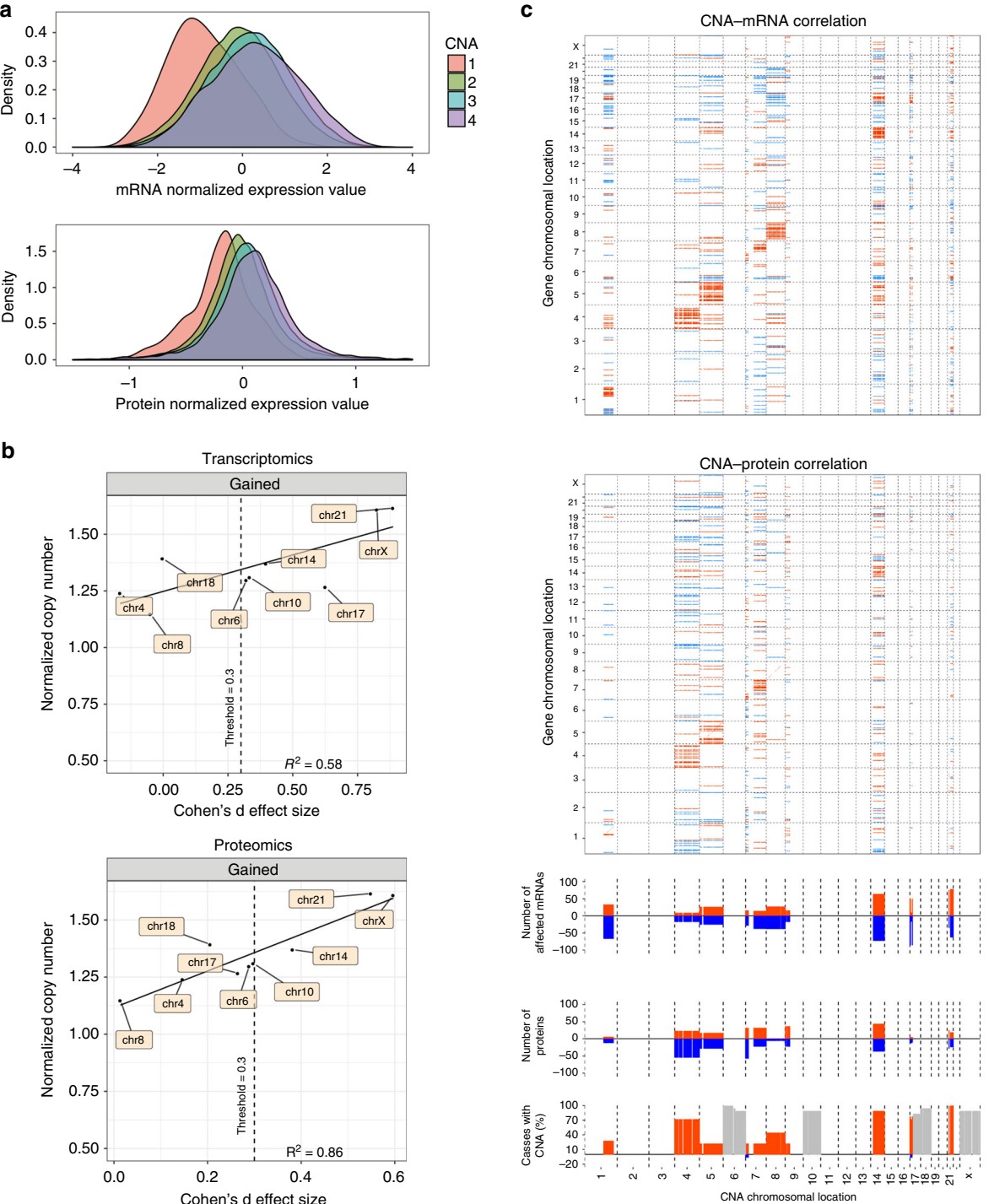

**Fig. 2** Effects of copy number alterations on mRNA and protein abundance. **a** Dosage effects in 18 high hyperdiploid ALL at the RNA and protein levels. The effects were lower on the protein level than on RNA level, showing additional layers of control for protein expression. **b** Cohen's d effect size analysis of gained chromosomes in high hyperdiploid vs. *ETV6/RUNX1*-positive leukemia. **c** *cis* and *trans* effects of copy number changes in 18 cases of hyperdiploid childhood B-cell precursor acute lymphoblastic leukemia. Correlations of copy number aberrations (CNA) (*x*-axes) to RNA (top) and protein (bottom) expression levels (*y*-axes) are shown. Note that a large fraction of the genome was not included in the analysis since there was no copy number variance, either because all cases had two copies or because all cases had three copies. Significant (multiple-test adjusted $P < 0.05$) positive (red) and negative (blue) correlations between CNA and mRNAs/proteins are indicated. CNA *cis* effects appear as a red diagonal line, CNA *trans* effects as vertical stripes. The fraction (%) of significant CNA *trans* effects (positive in red and negative in blue) for each CNA gene is shown below. The bottom panel shows the fraction (%) of leukemias harboring CNA (copy number gain in red and copy number loss in blue). Chromosomes that were gained in more than 16 cases were not informative; their copy number is shown in gray

level, presumably because of feedback loops controlling expression and protein turnover. Cohen's d effect size analysis showed that gain of chromosomes X, 14, and 21 was associated with stronger dosage effects compared with the other commonly gained chromosomes in both the RNA-seq and proteomics datasets, with a linear relationship between normalized copy number and effect size (Fig. 2b). Thus, our data clearly support previous studies showing a general—albeit not ubiquitous—upregulation of genes in high hyperdiploid ALL[3,6]. Furthermore, we demonstrate that these dosage effects also are seen at the protein level, with proteins encoded on the gained chromosomes generally being more highly expressed.

To further investigate *cis* as well as *trans* (genes/proteins in other genomic regions) effects of copy number changes, we used a linear regression model to study the correlation between copy number and expression (Fig. 2c and Supplementary Fig. 3)[18]. In addition to the MS and RiboZero RNA-seq datasets, RNA expression from a previously published RNA-seq study was analyzed, comprising 83 cases (oligo(dT) RNA-seq; European Genome-phenome Archive accession number EGAD00001002112; Supplementary Data 1 and 6)[19]. In order to avoid outlier-driven results, only 2080 genes displaying copy number variation involving more than three cases were retained in the *cis*-effect analysis. Again, *cis* dosage effects were seen, involving 25% (524/2080) of genes and 12% (245/2080) of proteins at a significance level of $P < 0.05$ (Fig. 2c, Supplementary Fig. 3). Furthermore, *trans* effects on the whole transcriptome and proteome were seen for all informative regions. These were generally lower in the proteome data, in particular for the long arm of chromosome 1 and chromosomes 8, 17, and 21, which displayed very few *trans* effects in the protein dataset. To further investigate the leukemogenic impact of individual chromosomal gains, we mapped known cancer driver genes to see whether they were associated with the chromosomal pattern of high hyperdiploid ALL. That is, whether oncogenes were more and tumor suppressor genes less commonly located on the frequently gained chromosomes. However, no such association was seen (Supplementary Fig. 3). Taken together, the analysis of copy number and gene/protein expression confirmed that the extra chromosomes in high hyperdiploid ALL have a large impact at RNA and protein levels in both *cis* and *trans*.

**Protein expression differences between leukemic subtypes.** Next, we focused on expression differences between high hyperdiploid and *ETV6/RUNX1*-positive ALL. To investigate whether proteomics could be used to distinguish between high hyperdiploid and *ETV6/RUNX1*-positive leukemia, hierarchical cluster and principal component analyses were performed. The two subtypes clustered separately in unsupervised analyses, both by RNA and protein expression, in 27 cases (Fig. 3a). In supervised analysis, 2423 genes and 1286 proteins were upregulated and 2222 genes and 1127 proteins were downregulated in high hyperdiploid cases compared with *ETV6/RUNX1*-positive cases (multiple-test adjusted $P \leq 0.05$) (Supplementary Data 4 and 5). Of these, 684 upregulated and 624 downregulated genes and proteins overlapped (Supplementary Data 4 and 5). Overall, there was a linear relationship between the log2 fold changes of RNA-seq and proteomics data (Spearman's correlation coefficient = 0.54, $P < 2.2e-16$) (Fig. 3b), with the correlation being stronger for gene/protein pairs with high fold changes.

Several of the top differentially expressed proteins have previously been reported to play a role in leukemogenesis or to be associated with ALL. These include, for example, CD44 and FLT3 (Supplementary Data 4 and Supplementary Fig. 4), which were highly expressed in high hyperdiploid cases. *ETV6/RUNX1*-

positive BCP-ALL has previously been reported to display a CD44[low-negative] immunophenotype[20], agreeing well with this protein being differentially expressed by proteomics. In regards to FLT3, the *FLT3* gene harbors activating mutations in approximately 10–20% of high hyperdiploid ALL and has previously been reported to be highly expressed in high hyperdiploid ALL regardless of mutational status[3,21]. Here we show that this high expression is maintained at the protein level, suggesting that FLT3 may be involved in the leukemogenesis of high hyperdiploid childhood ALL also in the absence of mutations. A comparison with six sorted pro-B/pre-B samples—the normal cells considered to be closest to ALL blasts—in the oligo(dT) RNA-seq dataset confirmed that *CD44* and *FLT3* were highly expressed in the hyperdiploid leukemias (Supplementary Fig. 4). Among the top-downregulated proteins in high hyperdiploid cases were IGF2BP1, CLIC5, RAG1, and RAG2, which also showed low RNA expression compared with the normal pro-B/pre-B dataset (Supplementary Data 4 and Supplementary Fig. 4). *IGF2BP1* is recurrently involved in fusions with *IGH@* in BCP ALL and has previously been reported to be highly expressed in *ETV6/RUNX1*-positive cases[22,23]. *CLIC5* has been shown to be a target of ETV6 and loss of ETV6 leads to its upregulation, providing the cells with higher resistance to lysosome-mediated apoptosis[24]. As regards RAG1 and RAG2, these are key components of somatic V(D)J recombination and this process has previously been shown to be involved in *ETV6/RUNX1*-mediated leukemogenesis[25], agreeing well with the high expression seen in our cohort. Taken together, the top differentially expressed proteins obtained by MS agree well with previously reported RNA expression results, supporting the validity of our proteomics approach.

Gene set enrichment analysis (GSEA) was performed to identify dysregulated pathways in BCP-ALL. Pathways that were enriched in high hyperdiploid ALL in the protein analysis could be divided into six different categories: (1) translation and ribosomes, (2) innate immunity, (3) cell adhesion, (4) cytokines and activated signaling, (5) protein folding and proteolysis, and (6) the endosome (Fig. 3c and Supplementary Data 7). Pathways that were enriched in *ETV6/RUNX1*-positive cases comprised those related to: (1) chromatin organization, modification, and structure, (2) the G2/M checkpoint, and (3) mitochondria (Fig. 3c and Supplementary Data 8). Support for enrichment at the RNA level was seen for all these processes expect pathways related to mitochondria, both in the RiboZero and the oligo(dT) datasets (Supplementary Data 9–12). Taken together, the GSEA results suggest an upregulation of translation and protein metabolism, including proteolysis and the endosome, in high hyperdiploid ALL. This may be explained by the additional transcription from the extra chromosomes, which would be expected to result in a general increase in translation. Furthermore, aneuploidy, in particular hyperdiploidy, has been reported to be associated with a proteotoxic stress response related to increased strain on the protein folding pathways of the cell[26], which could explain the enrichment for protein folding and proteolysis seen here. The relative downregulation of pathways related to chromatin organization, modifications and structure may be related to a higher proliferative capacity of *ETV6/RUNX1*-positive cases, but could also be associated with epigenetic events in high hyperdiploid ALL, in particular in light of the changes in chromatin organization that we found by high-resolution chromosome conformation capture (Hi-C) in this subtype (see below). That the G2/M checkpoint is enriched in *ETV6/RUNX1*-positive cases, on the other hand, agrees well with the previously reported importance of DNA recombination in such cases[25].

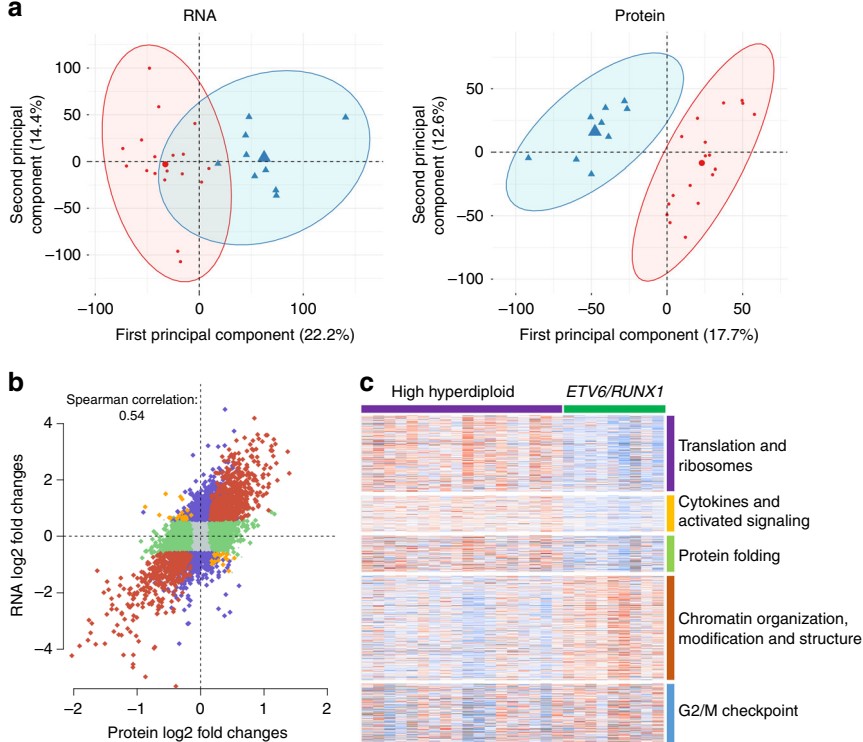

**Fig. 3** Clustering and enriched pathways in high hyperdiploid and *ETV6/RUNX1*-positive leukemia. **a** Principal component analyses of 27 B-cell precursor acute lymphoblastic leukemias showed that high hyperdiploid (red) and *ETV6/RUNX1*-positive (blue) cases clustered separately in unsupervised analyses by scaled RNA (left) and protein (right) abundance. **b** There was a linear relationship between the log2 fold changes of RNA-sequencing and proteomics data (Spearman's correlation coefficient = 0.54) between the high hyperdiploid and *ETV6/RUNX1*-positive subtypes, demonstrating a high correlation between changes on the transcript and translation levels of an individual gene product. The correlation was stronger for gene/protein pairs with high fold changes. Significant changes found in both RNA-seq and LC-MS/MS are shown in red, inverse changes found in RNA-seq and LC-MS/MS in yellow, significant changes found only in LC-MS/MS in green, and changes only found in RNA-seq in purple. **c** Gene set enrichment analysis of protein data highlighted sets of pathways that were significantly different between high hyperdiploid and *ETV6/RUNX1*-positive cases

**Transcriptional dysregulation in high hyperdiploid ALL.** We further found that CTCF, as well as several members of the cohesin complex, were significantly lower expressed at both the RNA and protein levels in our high hyperdiploid cases compared with *ETV6/RUNX1*-positive cases as well as compared with normal pro-B/pre-B cells (Fig. 4a, Supplementary Fig. 5). Analyses of the oligo(dT) RNA-seq dataset and two different publicly available array-based gene expression datasets (GEO accession numbers GSE13351 and GSE13425) confirmed that the expression of both CTCF and cohesin seems to be particularly low in high hyperdiploid ALL compared with other types of childhood ALL (Supplementary Fig. 5). We did not observe any differences in complex formation or correlation between the cohesin complex members between the high hyperdiploid and the *ETV6/RUNX1*-positive cases (Supplementary Fig. 5), indicating that there was a general downregulation of gene/protein expression and not a disturbance of the complex formation. *CTCF* has recently been identified as a putative tumor suppressor gene in ALL and we have previously reported a *CTCF/PARD6A* fusion that presumably results in disruption of the normal function of CTCF in one case of high hyperdiploid ALL[3,27]. Besides being a transcription factor, CTCF binds to chromatin at interphase and, together with the cohesin complex, forms the basis for the formation of topologically associating domains (TADs); chromatin loops <1 Mb in size containing DNA sequences that interact more frequently with each other than with external sequences[28]. TADs are generally conserved between different tissues and their disruption leads to changes in gene expression when the insulating

function of the TAD boundaries is lost[28]. Thus, CTCF and cohesin are master regulators of transcription.

We hypothesized that this low expression of CTCF and cohesin could have genome-wide effects on the transcriptional regulation in high hyperdiploid ALL. To address this, we first investigated whether the differences in gene expression between high hyperdiploid and *ETV6/RUNX1*-positive ALL were associated with the number of CTCF binding sites in gene bodies and the flanking 5 kb, i.e., genes regulated by CTCF binding. We found that differentially expressed genes in both the oligo(dT) and RiboZero RNA-seq datasets were strongly enriched for more CTCF binding sites compared with genes that showed similar expression in the two ALL subtypes (chi-square test; $P = 3.41e-05$ and $P = 1.549e-05$, respectively; Supplementary Fig. 6), in line with the previous experimental data from a mouse model with reduced CTCF expression[29]. Furthermore, genes with higher numbers of CTCF binding sites showed larger fold changes in both datasets (Supplementary Fig. 6). We also classified genes as anchor genes or background genes based on the distance of their transcription start sites to the closest CTCF/cohesin anchors, forming the basis for chromatin loops, according to published chromatin interaction analysis with paired-end tag sequencing (ChIA-PET) data[30]. We found that a significantly higher proportion of the genes that were differentially expressed between high hyperdiploid and *ETV6/RUNX1*-positive leukemias were anchor genes in both the oligo(dT) and the RiboZero RNA-seq datasets (hypergeometric test; $P = 0.0139$ and $P = 0.00513$, respectively; Supplementary Fig. 7). Furthermore, differentially

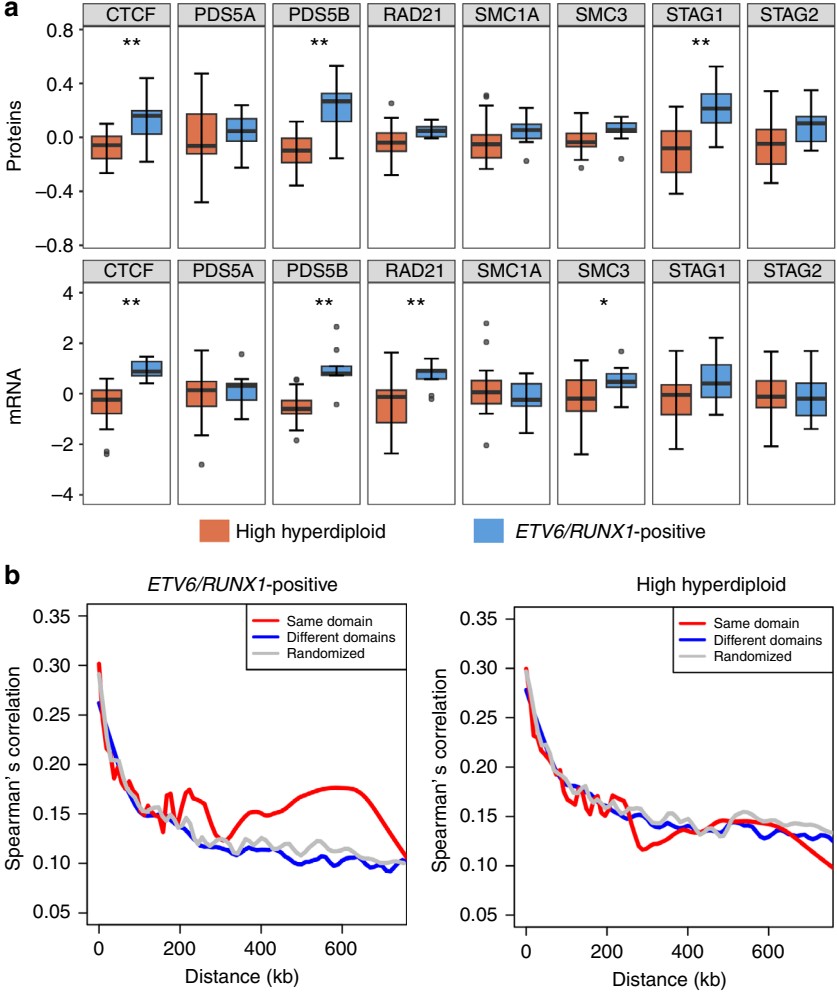

**Fig. 4** Low CTCF/cohesin expression and transcriptional dysregulation in high hyperdiploid leukemia. **a** Boxplots of the expression of CTCF and members of the cohesin complex in proteomics (top) and RiboZero RNA-sequencing datasets (bottom). Low expression of CTCF/cohesin complex members was seen in the high hyperdiploid subgroup at both the RNA and protein levels. The center of the boxplot is the median and lower/upper hinges correspond to the first/third quartiles; whiskers are 1.5 times the interquartile range and data beyond this range are plotted as individual points. **b** Spearman's correlation coefficient between gene pairs as a function of distance across the oligo(dT) RNA-sequencing dataset for *ETV6/RUNX1*-positive cases (left; $n = 39$) and high hyperdiploid ALL (right; $n = 44$). The analysis showed that the expression of gene pairs in the same topologically associating domain (TAD; red) displayed higher correlation than those in different domains (blue) or randomly selected regions (gray) in *ETV6/RUNX1*-positive cases, whereas no difference was seen in high hyperdiploid ALL, suggesting that transcriptional dysregulation in hyperdiploid cases is related to TAD borders

expressed anchor genes showed significantly higher fold changes (Mann–Whitney *U*-test; $P = 6.6e-6$ and $P = 1.31e-4$, respectively; Supplementary Fig. 7), suggesting that a portion of differentially expressed genes between hyperdiploid and *ETV6/RUNX1*-positive cases were the result of changes in CTCF binding.

To further investigate how the low levels of CTCF and cohesin affected genome-wide transcription, we used publicly available data to classify gene pairs according to whether they should be divided by a TAD boundary or not, since the overall TAD structure is generally conserved in human tissues[31]. We then investigated whether their expression was correlated. First, we analyzed RNA-seq data from a large cohort of childhood ALL ($n = 201$) including all genetic subtypes[19], from normal bone marrow ($n = 20$)[19], from acute myeloid leukemia (TCGA-LAML, https://portal.gdc.cancer.gov/projects/TCGA-LAML; $n = 151$)[32] and from papillary renal-cell carcinoma (TCGA-KIRP, https://portal.gdc.cancer.gov/projects/TCGA-KIRP; $n = 270$)[33]. These datasets all clearly displayed higher correlation between the

expression of gene pairs within the same TAD compared with gene pairs separated by a TAD boundary (Supplementary Fig. 8), showing that the TAD structure used in the analysis corresponded well with actual transcriptional regulation and was in line with previous studies[34]. We then performed the same analysis in high hyperdiploid ($n = 44$) and *ETV6/RUNX1*-positive ($n = 39$) cases separately. Whereas the *ETV6/RUNX1*-positive leukemias showed a difference between intra- and inter-TAD gene pairs, similar to the other datasets investigated, no correlation with the expected TAD structure was observed in the high hyperdiploid samples (Fig. 4b). Analysis of two publicly available array-based gene expression datasets from childhood ALL (GEO accession numbers GSE13351 and GSE13425) confirmed that high hyperdiploid ALL displayed aberrant expression in relation to the expected TAD structure (Supplementary Fig. 8). Restricting the analysis to only the commonly trisomic or only the commonly disomic chromosomes did not change the result (Supplementary Fig. 8), suggesting that the phenomenon is not directly associated with the copy number of

individual chromosomes. Taken together, the analyses suggest that high hyperdiploid ALL exhibits an aberrant gene expression pattern associated with changes in DNA looping.

**TAD boundaries in high hyperdiploid ALL**. To investigate further the TAD organization in childhood ALL, we performed in situ Hi-C analysis on four high hyperdiploid and two *ETV6/ RUNX1*-positive cases (Supplementary Data 13). Raw sequencing data were processed using the HiCUP pipeline[35], resulting in 230–320 million unique valid sequence tags per sample. An average of 3450 TADs (range 2965–3960) was identified per case by Domaincaller at 25 kb resolution with a mean size of ~740 kb (range 645–845 kb) (Supplementary Data 13). The average number of boundaries was 4230 (range 3853–4659) (Supplementary Data 13). The majority of TAD boundaries were expected to be bound by the insulator protein CTCF (86%) and the cohesin subunit RAD21 (80%), in line with previous reports (Supplementary Data 13)[31]. Comparing the TAD boundaries of our samples with the high resolution Hi-C dataset from the human lymphoblastoid cell line GM12878[31] showed that approximately 70% of the TAD boundaries we found were also present in GM12878, indicating that the overall TAD structure was intact in the leukemia samples (Fig. 5, Supplementary Data 13).

We then investigated whether the low expression of CTCF/ cohesin affects higher-order segregation of active and inactive chromosome domains into A and B compartments. We determined the compartment types of the genome at 500 kb resolution in leukemia samples as well as GM12878 cell line using Juicer eigenvector[36]. Overall, most (>90%) genomic regions were in the same compartment in the leukemia samples as in the GM12878 cell line and more than 98% of genomic regions were in the same compartment in high hyperdiploid ALL and *ETV6/RUNX1*-positive cases, in line with previous studies showing that depletion of CTCF does not lead to compartment switching[37].

Comparing *ETV6/RUNX1*-positive and high hyperdiploid cases, the number of TAD boundaries were reduced and the average TAD structure length was increased by 21–120 kb in three of four high hyperdiploid ALL. The exception (case HeH_42) had an average TAD length of 645 kb (Supplementary Data 13). This increase in TAD lengths in the three hyperdiploid cases resulted from the partial fusion of multiple TADs into one, in line with the decrease in the number of TAD boundaries. In order to analyze changes in chromatin organization, we focused on recurrent changes in boundaries detected in the different subgroups of leukemia. One hundred thirty-one boundaries were weakened or absent in at least two high hyperdiploid samples whereas only 14 boundaries were absent in both *ETV6/RUNX1*-positive cases (Fig. 5 and Supplementary Data 14). Most of the corresponding boundaries overlapped CTCF-cohesin binding sites (127/131, 97%), indicating that the loss of TAD boundaries was associated with loss of a functional CTCF/cohesin complex in high hyperdiploid samples (Supplementary Data 14). We then checked the expression of mRNA and proteins in the RiboZero RNA-seq and MS datasets. Of the 298 expressed mRNAs and 210 expressed proteins encoded within 1 Mb of the lost boundaries, significant (multiple-test adjusted $P \leq 0.05$) expression differences between high hyperdiploid and *ETV6/RUNX1*-positive ALL could be seen for 134 (45%) and 65 (31%), respectively (Supplementary Data 15). Of these differentially expressed genes/proteins, 98/134 (73%) genes and 42/64 (66%) proteins were downregulated in high hyperdiploid samples, which was more often than by chance (chi-square test, $P = 4.6\mathrm{e}{-9}$ for RNA-seq, and $P = 0.0032$ for proteomics). This indicates that changes in chromatin organization caused by CTCF/cohesin complex depletion tend to downregulate gene expression in high hyperdiploid cases.

Although the global chromosomal interaction pattern appeared largely unchanged between high hyperdiploid, *ETV6/RUNX1*-positive cases and the cell line GM12878, closer inspection showed differences in the strength of internal interactions within TADs. To test whether the TAD interaction strength was affected, we used directionality index and insulation score analyses to calculate the ratio of interactions found within TADs versus those spanning a boundary[38,39]. We found a genome-wide change in directionality index as well as in insulation score between cases: two high hyperdiploid cases (HeH_9 and HeH_10) showed a clear loss of boundary strength based on directionality index analysis and three (HeH_9, HeH_10, and HeH_48) showed reduced insulation based on insulation score analysis compared with the GM12878 cell line (Fig. 6). Thus, whereas the position of TAD boundaries remained largely unchanged in high hyperdiploid ALL samples, their quality was affected by changes in local and distal interactions, with a fraction of TADs losing insulation strength. This suggests that at least a subset of high hyperdiploid ALL have significant loss of insulation at TAD borders, agreeing well with the observed transcriptional dysregulation in this subgroup.

**Poor metaphase chromosome morphology in hyperdiploid ALL**. To further explore the possibility of an aberrant chromatin organization in high hyperdiploid ALL, we next focused on the metaphase chromosomes. Metaphase chromosome morphology, corresponding to the number of bands obtained with banding techniques as well as the size and general appearance of the chromosomes, varies between different cells and tissues. CTCF is bound to chromatin throughout the cell cycle and it has been suggested that it also affects the metaphase chromosome architecture[40]. We, therefore, hypothesized that high hyperdiploid ALL may have aberrant chromosome morphology. In fact, a common opinion among hematological cytogeneticists is that this genetic subtype displays particularly poor chromosome morphology, although this has not, to the best of our knowledge, been properly investigated. To address this issue, we developed a scale from 1 to 3 (1 corresponding to poor and 3 to good morphology; Fig. 7) for scoring chromosome morphology (chromosome morphology score; CMS) in a consistent manner and applied it to 37 cases of high hyperdiploid ALL and 33 cases of *ETV6/RUNX1*-positive ALL. Although the CMS varied between cells within the same case, there were clear differences in the mean values between cases (Supplementary Data 16). Furthermore, the investigation revealed a difference in mean CMS between the two genetic subtypes, with high hyperdiploid ALL displaying significantly lower CMS, corresponding to poorer chromosome morphology (Mann–Whitney one-sided test; $P = 0.0075$; mean CMS 1.8 vs. 2.1; Fig. 7; Supplementary Data 16). Thus, metaphase chromosomes of high hyperdiploid ALL show signs of an aberrant chromatin organization, in line with our RNA-seq and Hi-C results.

## Discussion

We here report a full-scale proteogenomic analysis of childhood ALL, including the two largest subtypes of this disease that together constitutes more than half of cases. The investigation encompassed more than 8000 proteins and 12,000 RNAs in genetically well-characterized cases. To the best of our knowledge, only primary tumor samples from rhabdomyosarcoma, colon and rectal, prostate, and breast cancer have previously been subjected to proteogenomic analyses to this level[9–11,41]. Although many previous studies of childhood ALL have utilized RNA expression,

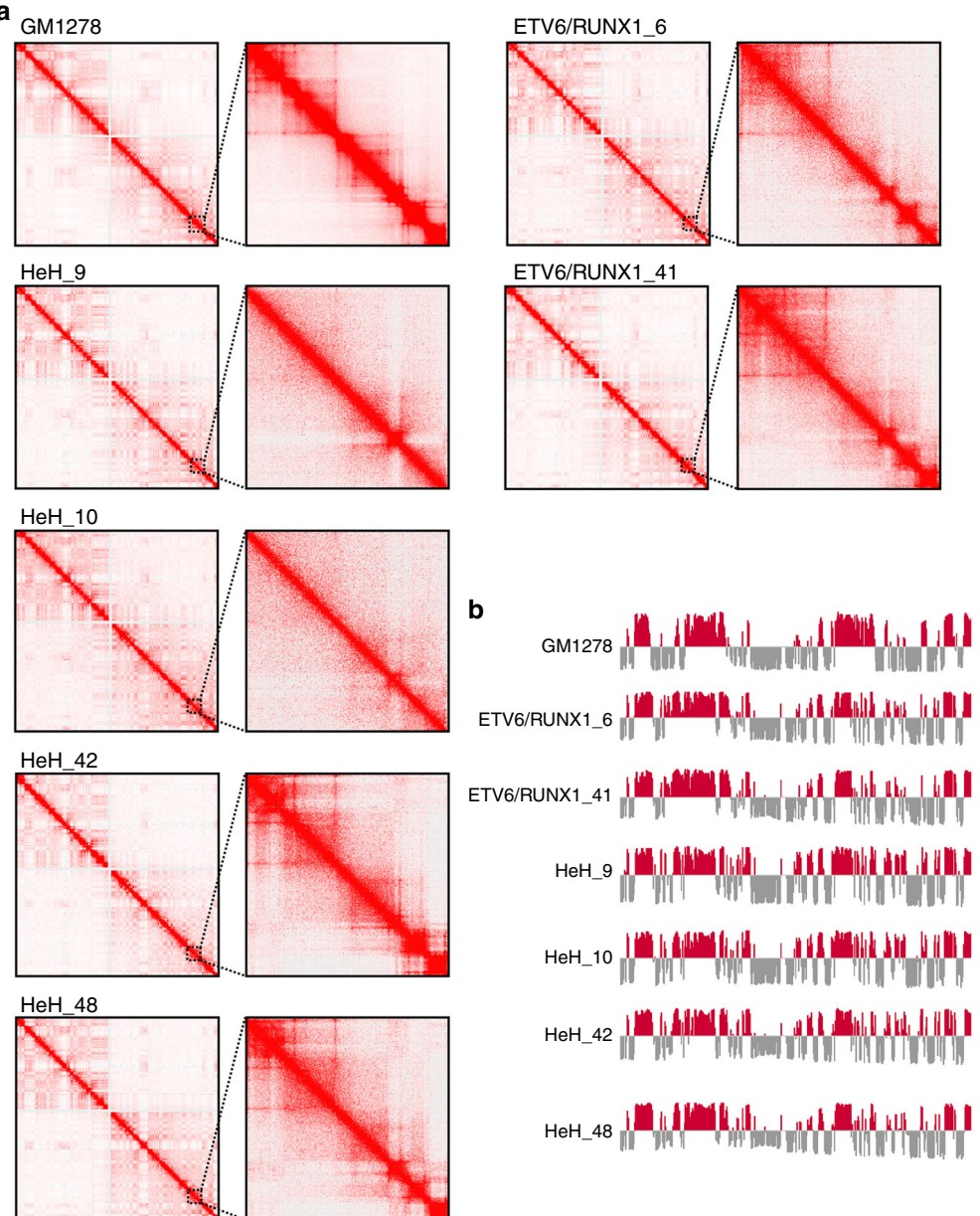

**Fig. 5** Hi-C of high hyperdiploid and *ETV6/RUNX1*-positive cases. **a** Contact matrices from chromosome 3, selected because it is disomic in all cases and displays no structural aberrations. The whole chromosome at 250 kb resolution is shown to the left and the 161–172 Mb region at 25 kb resolution to the right. At a resolution of 250 kb, the interaction profile is similar, showing that the general chromatin architecture is intact. However, at a resolution of 25 kb, it can clearly be seen that two of the high hyperdiploid cases (HeH_9 and HeH_10) have lost some topologically associating domains. **b** A/B compartment profile of chromosome 3 in cell line GM12878 and the six leukemia samples at 500 kb resolution. The profiles were similar between the cell line and the leukemias

first with microarrays and more recently with RNA-seq[3,6,7], the protein levels are expected to have a more direct impact on the phenotype and our proteome analysis thus provides an improved insight into leukemogenesis.

Previous studies of RNA expression in high hyperdiploid ALL have shown clear dosage effects[3,6,7], i.e. a general upregulation of genes on the gained chromosomes, corresponding to *cis* effects. However, no data on the effect on protein expression or in-depth analyses of *cis* and *trans* effects in relation to the gained chromosomes have been published to date. Here, we show that the gained chromosomes in high hyperdiploid ALL are also associated with *cis* effects at the protein level, but that those effects are

weaker than at the RNA level. In addition, we also identified a general dysregulation of gene expression in relation to TAD boundaries in high hyperdiploid ALL. This corresponds to a likely disturbance in the insulation between regulatory elements and gene promoters that should be separated by a TAD boundary. Depleting CTCF experimentally has been associated with loss of insulation at TAD borders in a dose-dependent manner, whereas cohesin loss has been linked to lower insulation between TADs as well as depletion of TADs[37,42,43]. Thus, it is feasible that the relatively low expression of CTCF and cohesin that is seen in high hyperdiploid ALL could affect TAD border insulation and/or TAD structure. In line with this, our Hi-C analysis revealed that

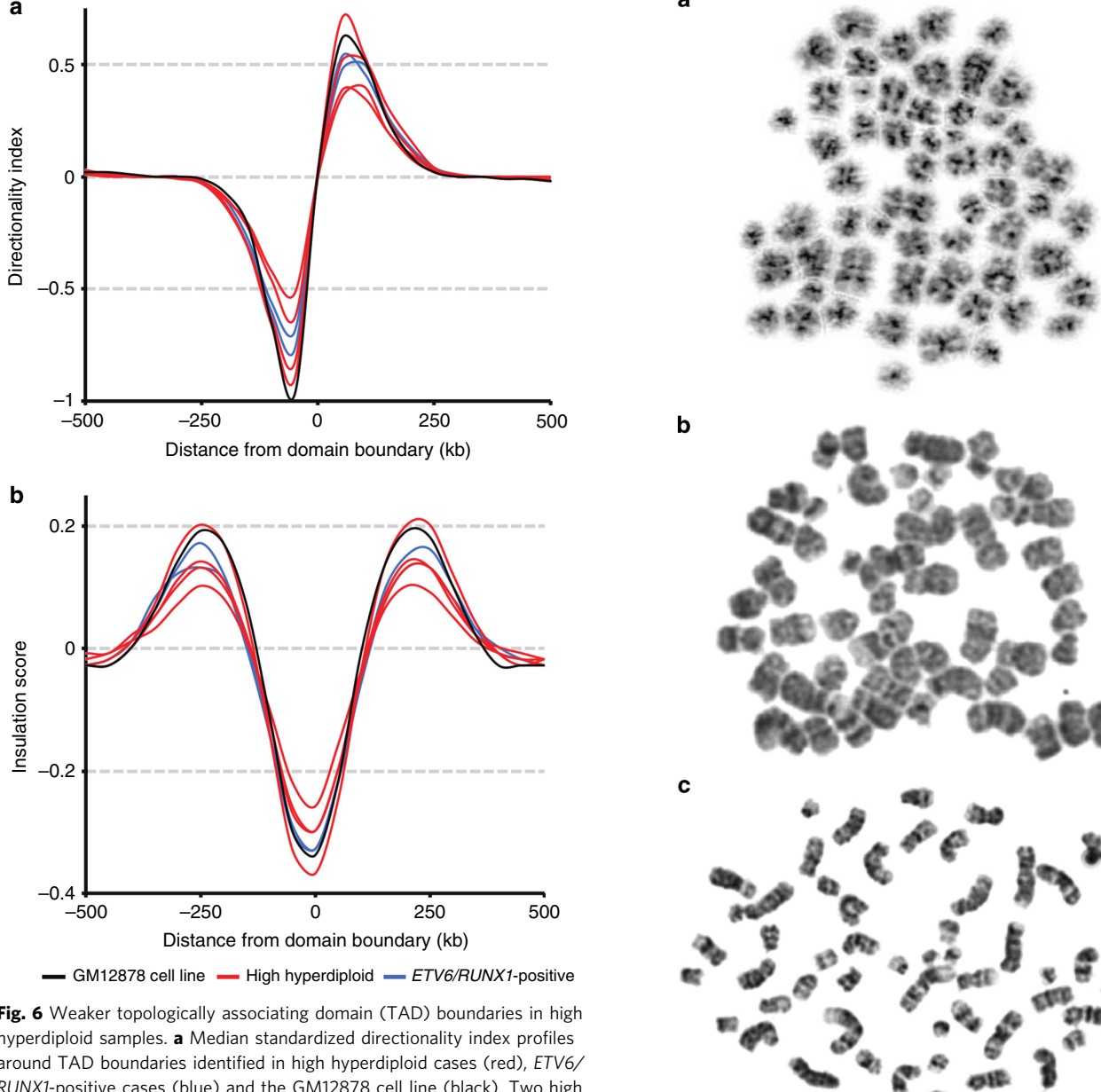

Fig. 6 Weaker topologically associating domain (TAD) boundaries in high hyperdiploid samples. **a** Median standardized directionality index profiles around TAD boundaries identified in high hyperdiploid cases (red), *ETV6/RUNX1*-positive cases (blue) and the GM12878 cell line (black). Two high hyperdiploid cases (HeH_9 and HeH_10) showed markedly decreased boundary strength, indicating permissive TAD boundaries. **b** Median insulation score around the TAD boundaries identified in high hyperdiploid cases (red), *ETV6/RUNX1*-positive cases (blue) and the GM12878 cell line (black). Three high hyperdiploid cases (HeH_9, HeH_10, and HeH_48) showed decreased insulation signal amplitude suggesting weaker insulation between TADs compared to the remaining samples

Fig. 7 Metaphase chromosome morphology in hyperdiploid leukemia. High hyperdiploid childhood acute lymphoblastic leukemia samples displayed varying metaphase chromosome morphology, but the majority of cases had poor morphology. **a** Example of metaphase with score 1—poor morphology (case HeH_33). **b** Example of metaphase with score 2—fair morphology (case HeH_48). **c** Example of metaphase with score 3—good morphology (case HeH_48)

three of four investigated high hyperdiploid samples displayed weaker TAD boundaries than the remaining leukemias and control cell lines, although the relatively small number of samples investigated prevents definite conclusions. Additionally, a higher number of TAD boundaries were recurrently lost in high hyperdiploid cases compared with the *ETV6/RUNX1*-positive cases and the cell lines. Furthermore, we found that high hyperdiploid ALL display an aberrant chromosome metaphase morphology, also suggesting an aberrant chromatin architecture. The underlying cause of the low CTCF and cohesin expression in high hyperdiploid ALL is currently unknown but mutations are unlikely to be the general cause; although these do occur, the

frequency is relatively low[3]. *CTCF* is encoded on chromosome 16, which is rarely gained[2], whereas the core members of the cohesin complex are all encoded on commonly gained chromosomes (8q24 for *RAD21*, 10q25 for *SMC3*, Xp11 for *SMC1A* and Xq25 for *STAG2*); thus, the specific aneuploidy in high hyperdiploid ALL may cause relatively low expression of CTCF but not of cohesin.

Taken together, we show that the chromosomal gains in high hyperdiploid ALL are associated with genome-wide effects on

transcription. Furthermore, we show genome-wide transcriptional dysregulation with putative leukemogenic effects in relation to TAD borders in the hyperdiploid subtype suggestive of changes in chromatin architecture; such changes could also be seen by Hi-C in a subset of cases. Whether aberrant chromatin architecture is a common phenomenon in aneuploid tumors remains an open question.

## Methods

**Patients.** The study comprised a total of 48 high hyperdiploid and 41 *ETV6/RUNX1*-positive pediatric BCP-ALL cases that had been treated at Skåne University Hospital, Lund, Sweden, selected based on samples being available from diagnosis (Supplementary Data 1). The cohort included 52 boys and 37 girls, with a median age at diagnosis of 4 years (range 0–16) and a median white blood cell count of $7.8 \times 10^9/l$ (range 0.9–164). All cases had been tested for *BCR/ABL1*, *ETV6/RUNX1*, *PBX1/TCF3* and *KMT2A* (previously *MLL*) rearrangements by reverse-transcriptase PCR, fluorescence in situ hybridization, or Southern blot as part of the clinical analyses and were found to be negative for these fusion genes, with the exception of the *ETV6/RUNX1* fusion in that subgroup. Informed consent was obtained according to the Declaration of Helsinki and the study was approved by the Ethics Committee of Lund University.

**DNA and RNA extraction.** Details for samples subjected to oligo(dT)-based RNA-seq have been published elsewhere[19]. For samples subjected to mass spectrometry analysis, DNA, total RNA and proteins were extracted from bone marrow or peripheral blood samples obtained at diagnosis and stored in TRIzol (Thermo-Fisher Scientific, Waltham, MA) at −80 °C for 4–17 years. After addition of chloroform, RNA and DNA were precipitated according to the manufacturer's instructions. The remaining fractions containing proteins were stored at −80 °C.

**Sample preparation for mass spectrometry.** The stored TRIzol fractions were thawed and loaded into Slide-A-lyzer cassettes (ThermoFisher Scientific, 3.5-kDa cut-off, cat no 87722) and dialyzed according to the manufacturer's instructions against an aqueous solution containing 0.25% SDS and 25 mM Hepes pH 7.6 overnight at 4 °C with two changes of the dialysis buffer. The dialyzed samples were digested by a modified FASP-protocol[5]. Protein concentrations were estimated by gel staining and approximately 250 μg of each sample was mixed with 1 mM DTT, 8 M urea, 25 mM HEPES, pH 7.6 and transferred to a 10-kDa cut-off centrifugation filtering unit (Pall, Nanosep®, Merck, Darmstadt, Germany), and centrifuged at 14,000 × g for 15 min. Proteins were alkylated by 50 mM iodoacetamide (IAA) in 8 M urea, 25 mM HEPES for 10 min. The proteins were then centrifuged at 14,000 × g for 15 min followed by two more additions and centrifugations with 8 M urea, 25 mM HEPES. Proteins were digested at 37 °C with gentle shaking overnight by addition of Lys-C (enzyme:protein = 1:50, Wako Pure Chemical Industries, Ltd.) in 500 mM Urea, 50 mM HEPES pH 7.6 followed by an additional overnight digestion with trypsin (enzyme:protein = 1:50, ThermoFisher Scientific) in 50 mM HEPES, pH 7.6. The filter units were centrifuged at 14,000 × g for 15 min followed by another centrifugation with MilliQ water and the flow-through was collected. Peptides were cleaned up by a modified sp3-protocol[44]. Briefly, carboxylate-modified paramagnetic beads (ThermoFisher Scientific; CAT No. 09-981-121 and ThermoFisher Scientific; CAT No. 09-981-123) were mixed 1:1 and washed with MilliQ water. 10 μl of the bead-mixture was added to each peptide sample. Acetonitrile was added so that the final concentration was >95% and beads were incubated at room temperature for 8 minutes. Next, beads were placed on a magnetic rack, the supernatant discarded and the beads washed twice with 180 μl of acetonitrile. Beads were re-suspended in 100 μl of MilliQ water and sonicated to release the peptides, supernatants were collected and stored at −20 °C. Peptide concentration was determined by the Bio-Rad DCC assay and 30 μg of peptides from each digested sample was labeled with TMT 10-plex reagent according to the manufacturer's protocol (ThermoFisher Scientific). A small portion of unlabeled peptides were pooled from all samples to generate an internal standard that was labeled with TMT-channel 131 and included in all sets. Labeled samples were pooled, cleaned by strata-X-C-cartridges (Phenomenex, Torrance, CA) and dried in a Speed-Vac.

**Peptide level sample fractionation through HiRIEF.** The TMT labeled peptides, 300 μg, were separated by immobilized pH gradient - isoelectric focusing (IPG-IEF) on pH 3–10 strips using the HiRIEF method[45]. Peptides were extracted from the strips by a prototype liquid handling robot, supplied by GE Healthcare Bio-Sciences AB. A plastic device with 72 wells was put onto each strip and 50 μl of MilliQ water was added to each well. After 30 minutes incubation, the liquid was transferred to a 96 well plate and the extraction was repeated two more times with 35% acetonitrile (ACN) and 35% ACN, 0.1% formic acid in MilliQ water, respectively. The extracted peptides were dried in Speed-Vac and dissolved in 3% ACN, 0.1 % formic acid.

**Mass spectrometry based quantitative proteomics.** Extracted peptide fractions were separated using an Ultimate 3000 RSLCnano system coupled to a Q Exactive (ThermoFisher Scientific). Samples were trapped on an Acclaim PepMap nanotrap column (C18, 3 μm, 100 Å, 75 μm × 20 mm, ThermoFisher Scientific), and separated on an Acclaim PepMap RSLC column (C18, 2 μm, 100 Å, 75 μm x 50 cm, ThermoFisher Scientific). Peptides were separated using a gradient of mobile phase A (5% DMSO, 0.1% FA) and B (90% ACN, 5% DMSO, 0.1% FA), ranging from 6 to 37 % B in 60 min (depending on IPG-IEF fraction complexity) with a flow of 0.25 μl/min. The Q Exactive was operated in a data-dependent manner, selecting top 10 precursors for fragmentation by HCD. The survey scan was performed at 70,000 resolution from 400–1600 m/z, with a max injection time of 100 ms and target of $1 \times 10^6$ ions. For generation of HCD fragmentation spectra, a max ion injection time of 140 ms and AGC of $1 \times 10^5$ were used before fragmentation at 30% normalized collision energy, 35,000 resolution. Precursors were isolated with a width of 2 m/z and put on the exclusion list for 70 s. Single and unassigned charge states were rejected from precursor selection.

**Peptide and protein identification.** Orbitrap raw MS/MS files were converted to mzML format using msConvert from the ProteoWizard tool suite[45]. Spectra were then searched using MSGF+ (v10072)[46] and Percolator (v2.08)[47], where search results from eight subsequent fraction were grouped for Percolator target/decoy analysis. All searches were done against the human protein subset of Ensembl 75 in the Galaxy platform. MSGF + settings included precursor mass tolerance of 10 ppm, fully-tryptic peptides, maximum peptide length of 50 amino acids and a maximum charge of 6. Fixed modifications were TMT-10plex on lysines and peptide N-termini, and carbamidomethylation on cysteine residues, a variable modification was used for oxidation on methionine residues. Quantification of TMT-10plex reporter ions was done using OpenMS project's IsobaricAnalyzer (v2.0)[48]. Peptide-spectrum matches (PSMs) found at 1% false discovery rate (FDR) were used to infer gene identities. Protein quantification by TMT 10-plex reporter ions was calculated using TMT PSM ratios to the entire sample set (all 10 TMT-channels) and normalized to the sample median. The median PSM TMT reporter ratio from peptides unique to a gene symbol was used for quantification. Protein false discovery rates were calculated using the picked-FDR method using gene symbols as protein groups and limited to 1% FDR[49].

**DNA sequencing analyses.** WGS results for cases HeH_2, HeH_3, HeH_12, HeH_13, HeH_17, HeH_22, HeH_24, HeH_25, HeH_27, HeH_32, HeH_35, HeH_38, and HeH_43 and WES results for cases HeH_1, HeH_4, HeH_5, HeH_7, HeH_8, HeH_16, HeH_19, HeH_26, HeH_34, HeH_37, and HeH_41 have been previously published[3]. Briefly, for WGS, matched diagnostic and remission bone marrow or peripheral blood samples were sequenced to ~100x coverage on the Complete Genomics platform. Somatic events were identified using the Complete Genomics Cancer Sequencing v2.0 pipeline with CGA tools. For WES, libraries were constructed using the SureSelectXT2 Human All Exon V4 kit (Agilent Technologies, Santa Clara, CA) from matched diagnostic and remission bone marrow or peripheral blood samples and paired-end sequencing were done to ~120x coverage on an Illumina HiSeq2000. Somatic mutations were detected with MuTect[50]. WES for cases HeH_6, HeH_9-HeH_11, HeH_14, HeH_15, HeH_18, and ETV6/RUNX1_1-ETV6/RUNX1_9 were done by Nextera Rapid Capture Expanded Exome Kit (Illumina, San Diego, CA, USA), with ~110x coverage on an Illumina NextSeq 500. Paired remission samples were available from cases HeH_6, HeH_10, HeH15, ETV6/RUNX1_1-ETV6/RUNX1_5, and ETV6/RUNX7_1-ETV6/RUNX1_9. Pair-end sequence reads were aligned to the human_g1k_v37 by Burrows-Wheeler Aligner (BWA)[51]. Duplicate reads were marked with Picard and Indel realignment was performed with GATK[52]. Somatic mutations were identified using MuTect[50] and MuSE[53] whereas somatic indels were identified by manta[54] and strelka[55] with default settings. Mutations that passed the internal filters of the variation caller were further filtered by a minimum depth of 10 reads. For tumor samples without matched normal (HeH_9, HeH_11, HeH_14, HeH_18 and ETV6/RUNX1_6), variations were identified by GATK UnifiedGenotyper and annotation parameters QD (variant confidence/quality by depth) <2.0, MQ (RMS mapping quality) <40.0, FS (Fisher strand) 60.0, HaploTypeScore >13.0, MQRankSum < −12.5 and ReadPosRankSum <−8.0 were used to filter low quality variations. High-quality variants were further filtered by 1000 Genomes (20110521 release), ESP6500, ExAC, CG46 (popfreq_max_20150413) and 170 million variants (kaviar_20150923) provided by ANNOVAR[56] to remove potential SNP sites. Functional annotation was performed by ANNOVAR.

**SNP array analyses.** SNP array analysis was done on DNA extracted from diagnostic bone marrow or peripheral blood samples on the Human1M-Duo, Human-Omni1-Quad, Human-Omni5-4v (Illumina, San Diego, CA), or CytoScan HD platforms (Applied Biosystems, Thermo Fisher); data have been published previously[57].

**RNA-sequencing.** Details on the oligo(dT) RNA-seq dataset have been previously published[19]. Briefly, cDNA sequencing libraries were constructed from poly-A-selected RNA using the Truseq RNA library preparation kit v2 (Illumina, San Diego, CA) and sequenced on an Illumina HiScan SQ or an Illumina NextSeq 500.

For the RiboZero RNA-seq, RNA from cases HeH_1-HeH_18 and ETV6/RUNX1_1-ETV6/RUNX1_9 were constructed using the Human Ribo-Zero rRNA Removal Kit (Illumina, San Diego, CA) and sequenced on an Illumina NextSeq 500.

**Expression analyses**. RNA sequencing data were processed using the TCGA mRNA-seq pipeline (https://docs.gdc.cancer.gov/Data/Bioinformatics_Pipelines/Expression_mRNA_Pipeline/#mrna-analysis-pipeline). Briefly, sequencing reads were aligned to the human GRCh38 genome assembly using STAR[58] and read counts for each gene were obtained by HTSeq-count[59].

Genes with count-per-million (CPM) value greater than 1 were defined as expressed genes and only genes expressed in more than 80% of samples in at least one sample group were used for further analyses. For differential expression analysis, batch effects were adjusted by using RUVg function in RUVSeq[60] and differentially expressed genes were identified by using edgeR[61]. Benjamini–Hochberg adjusted (BH-adjusted) $P \leq 0.05$ were used as cutoff. In the oligo(dT) dataset, six samples representing sorted pro-B (CD34+CD38+CD19+CD10+) or pre-B (CD34−CD38+CD19+CD20−CD10+) cells were included and used to compare leukemic samples to the closest normal cell.

For the proteomics dataset, absolute intensity values of the PSMs were converted to ratios based on the pool reference and log2 transformed. Spectra mapping to unique gene symbols were retained and aggregated to proteins using the median value of the PSM ratios. Proteins identified in all sets were used for subsequent analysis. To remove the batch effects of proteomics data, iterative RUV4 algorithm was used[62]. In brief, we initialized the search with the median normalized dataset to detect the proteins with BH-adjusted $P$ greater than 0.9 as the first controls by using limma[63]. In each of the 10 iterations, we applied the RUV4 algorithm on the un-normalized data to obtain the residuals and calculate ab initio the control proteins as those with differential abundance of BH-adjusted $P$ greater than 0.9. These proteins were used as control proteins in the next iteration. Control proteins identified in the last iteration constituted the empirical controls which were uncorrelated to the tumor category. Protein differential expression analysis was performed by limma and BH-adjusted $P \leq 0.05$ was applied as cutoff for identifying differentially expressed proteins.

Two gene expression datasets obtained from NCBI Gene Expression Omnibus (accession numbers GSE13351 and GSE13425) were analyzed. The expression data were analyzed using Transcriptome Analysis Console (Affymetrix, Santa Clara, CA, USA) with default settings.

**Correlation between mRNA and protein variation**. To compare mRNA and protein variations across samples, we focused on 8222 genes/proteins that were detected in both the Ribozero RNA-seq dataset and the proteomics dataset. We first calculated the Spearman's correlation coefficient between RNA-seq FPKM values and RUV4-normalized values from the proteomics dataset across samples ($n = 27$) and $P$-values corresponding to the coefficients were computed and adjusted by Benjamini–Hochberg procedure. Significant calls were made based on BH-adjusted $P \leq 0.05$. Functional enrichment analysis was performed by GSEA-Pre-ranked algorithm[64] and Spearman's correlation coefficient was used as the ranking variable. Pairwise correlation analysis of protein pairs, which are present within the same complex of known protein complexes acquired from the CORUM database[65], were performed by using Spearman correlation's coefficient and the same analysis was performed on the RiboZero RNA-seq dataset. For the mRNA-protein stability analysis mRNA and protein half-lives from mouse fibroblast cell lines were extracted from Schwanhäusser et al.[13] and analyzed as per the original manuscript and Zhang et al.[9], i.e., stable (unstable) mRNAs and proteins were categorized according to their rank in the top (bottom) one third of half-lives, respectively. For micro-RNA targeting analysis miRNA-mRNA interactome data was downloaded from Helwak et al.[14]. For the analysis of the impact of ubiquitination and proteasomal degradation time-series ubiquitination data from HCT116 and 293T human cell lines upon bortezomib treatment was used[15]. We chose the 8-hour point as a proxy for steady-state and divided significantly/non-significantly ubiquitinated proteins according to an absolute log2 fold change greater/smaller than 1. We also investigated the impact of protein degradation profiles, namely exponential (ED) and non-exponential (NED) decay, using mouse fibroblast cell data from a click-chemistry assisted pulsed SILAC study[16]. Information on protein subcellular localization was downloaded from a MS-based study on global subcellular localization[17] and used to subset our dataset. For the analysis of the impact of differential expression/abundance on mRNA-protein correlations transcripts/proteins were divided to significant/non-significant based on BH adjusted $P$-value <0.05 and log2 fold changes higher (lower) than the 90% (10%) percentile. To avoid correlations being driven by tumor subtype differences (Simpson's paradox), partial correlations were estimated after regressing the data on tumor subtype and calculating Spearman correlations on the residuals. For the above analyses, we used overlapped gene symbols between datasets. Two-group and multi-group comparisons were assessed with two-sided Wilcoxon rank sum test and Kruskal–Wallis test, respectively.

**Analysis of *cis*- and *trans*-effects**. Matched copy number aberrations (CNA) based on SNP-array analysis, WES and/or WGS, proteomics and RNA-seq

measurements of 18 high hyperdiploid samples were used to study the impact of CNAs on mRNA and protein expression. In order to avoid outlier-driven results, only genes displaying CNAs involving more than 3 cases in each comparison group were retained (CNA genes, $n = 2080$). To analyze genome-wide *cis* effects of high hyperdiploid ALL samples, Spearman's correlation coefficient between genes/proteins abundance and copy number of 2080 informative CNA genes was calculated, respectively. To analyze genome-wide *trans* effects, the correlation between CNA genes and all 8222 mRNA and proteins detected in both RNA-seq and proteomics of 18 high hyperdiploid samples were determined using the MatrixeQTL R package[18]. Subsequently, $P$-values corresponding to the coefficient were calculated and significant CNA-mRNA and CNA-protein correlations were identified using BH-adjusted $P$-value 0.05 as cutoff.

To assess the impact of copy number aberrations on the hyperdiploid vs. *ETV6/RUNX1*-positive ALL differential expression, a normalized copy number per chromosome across the samples was calculated according to the formula:

$$Normalized\ copy\ number\ per\ chromosome$$
$$= Average_{samples}\left(\frac{\frac{\sum_{i=1}^{\#\ chromosomal\ segments} copy\ number\ (i) * length\ (i)\ in\ bp}{total\ length\ of\ chromosome}}{Normal\ ploidy\ (for\ somatic\ chromosomes-2, for\ sex\ chromosomes-1,2,NA)}\right) \quad (1)$$

Cohen's d effect size was calculated per chromosome at the mRNA and protein level and linearly regressed on the normalized copy number. We denoted significantly affected chromosomes as those with effect size >0.3.

To estimate the differential expression of known cancer driver genes[66] on the mRNA and protein level based on edgeR and limma, respectively, fold changes were overlaid on the ALL copy number landscape and gene symbols with BH-adjusted $P$-values ≤0.05 and fold change greater (lower) than the 90th (10th) percentile were displayed.

**Gene set enrichment analysis**. Gene set enrichment analysis[64] was done using the GSEA-pre-ranked algorithm with lists of all expressed genes ($n = 12,313$ for Ribozero and $n = 13,951$ for oligo(dT), respectively) and all expressed proteins ($n = 8480$), by using predictive log fold changes between high hyperdiploid ALL and *ETV6/RUNX1*-positive cases generated by edgeR (RiboZero and oligo(dT) RNA-seq) and log-fold changes values generated by limma (LC-MS/MS) as the ranking variable, respectively. We performed the analysis using the GSEA standalone software with default settings. Family-wise error rate (FWER) $P < 0.05$ was considered significant.

**CTCF binding site and ChIA-PET data analysis**. CTCF binding sites analysis was done according to Aitken et al.[29]. We downloaded the positions of CTCF binding sites from the ENCODE database (http://hgdownload.cse.ucsc.edu/goldenpath/hg19/encodeDCC/wgEncodeAwgTfbsUniform/wgEncodeAwgTfbsBroadGm12878CtcfUniPk.narrowPeak.gz) and the number of CTCF binding sites in each gene (plus 5 kb on either side) were obtained by using BEDTools (https://bedtools.readthedocs.io/en/latest/) command intersect. The proportion difference between the differentially expressed genes group and the remaining genes group was tested with the chi-square test and the difference between fold changes according to number of CTCF binding sites was tested by Mann–Whitney $U$-test. ChIA-PET data for CTCF and RAD21 were downloaded from the NCBI GEO database under accession numbers GSM1872886 and GSM1436265, respectively. To get high-confidence chromatin interactions, ChIA-PET interactions with low PET-count (less than ten reads coverage) were removed. BEDTools was used to find the overlapping interactions between CTCF and RAD21 ChIA-PET datasets and only interactions detected in both datasets were used. BEDTools command closest was used to determine the distance between the transcription start sites of expressed genes and CTCF/cohesin anchors. Genes located within 5 kb of a CTCF/cohesin anchor were defined as anchor genes. Statistical differences in the proportion of differentially expressed genes between the anchor genes group and the background group were tested by hypergeometric test and the differences between fold change values were tested with the Mann–Whitney $U$-test.

**Gene pair correlation analysis**. Gene pair correlation analysis was done according to Flavahan et al.[34]. Briefly, TADs of the IMR90 and GM12878 cell lines were downloaded from published Hi-C data[31] (Gene Expression Omnibus accession number GSE63525) and genes were assigned to the inner-most domain in which the transcription start site of the canonical transcript fell within. Genes were assigned to the same domain if they were assigned to the same domain in both GM12878 and IMR90 datasets. Ten thousand randomly generated domains were obtained by using BEDTools command random with 1 Mb as interval size. Spearman's correlation coefficient for all relevant gene pairs within the same TAD, different TADs and randomly generated domains were calculated and the correlation plot was smoothed by locally weighted scatterplot smoothing with weighted linear least squares (LOESS).

**Hi-C library preparation and sequencing**. Hi-C was done on cases HeH_9, HeH-10, HeH_42, HeH_48, ETV6/RUNX1_6, and ETV6/RUNX1_41, selected on the

basis of sample availability. Cell pellets approximately 5 mm in size containing mononuclear bone marrow or peripheral blood cells obtained at leukemia diagnosis were resuspended in 10 ml room temperature 1× PBS. The cells were fixed by the addition of 37% formaldehyde to a final concentration of 2% and gentle mixing on a rocker for 10 min at room temperature. The reaction was quenched by the addition of 1.5 ml cold glycine (0.125 M). Following incubation for 5 min at room temperature and 15 min on ice, the cells were pelleted by centrifugation at 400×g for 10 min at 4 °C. The pellet was resuspended in 1 ml cold 1× PBS by pipetting and made up to a final volume of 10 ml with cold 1× PBS. Finally, the cells were pelleted by centrifugation at 400×g for 10 min at 4 °C, and the pellet snap-frozen and stored at −80 °C until further analysis. In nucleus Hi-C on the crosslinked mononuclear cells was performed as outlined in Nagano et al.[67]. Each sample was split into two and processed separately to provide a technical replicate. One lane of 150 base pair paired-end sequencing was performed on the Illumina HiSeq 4000 instrument per replicate (12 lanes in all).

**Hi-C data analysis**. The sequences from the Hi-C libraries were mapped to reference human_g1k_v37 using HICUP with default settings[35]. Redundant reads and short-range Hi-C artifacts were removed from all downstream analyses. Filtered read pairs were then aggregated into 25, 50, and 100 kb genomic bins to generate Hi-C contact matrices. Low-coverage bins were filtered by using maximum allowed median absolute deviation (MAD-max) and low-coverage bins with MAD-max values higher than 2 were removed. Reads mapped to the same bin or adjacent bins were also removed. For the GM12878 cell line datasets, we downloaded.hic data from the NCBI GEO database (accession number GSE63525, GM12878_insitu_primary+replicate_combined_30.hic.gz) and converted.hic format into 25, 50, and 100 kb contact matrices by using the dump option of Juicertools[36]. The same filtering strategy was applied to the GM12878 cell line dataset. The filtered contact matrices were then normalized using the chromosome-adjusted iterative correction procedure (caICB) to eliminate copy number bias[68].

**TAD and boundaries calling**. The normalized 25 kb contact matrices of six leukemia cases and GM12878 cell line were used to predict TAD structures by DomainCaller[39], as this showed the best agreement with manual annotation in a previous study[69]. To find the optimal threshold for TAD calling, the window size parameter of DomainCaller was varied from 250 kb to 2 Mb and finally 500 kb window size was used, which showed about 80% of detected boundaries co-aligned with the previously detected boundaries of the GM12878 cell line[31] (accession number GSE63525, GM12878_primary+replicate_Arrowhead_domainlist.txt.gz) with ±100 kb precision. Standardized genome-wide directionality index value (z-score value) was used for TAD analysis. InsulationScore package was also used to identify TAD boundaries[38]. For insulation boundaries analysis, insulation square was set to 250 kb and insulation delta span was set to 125 kb. Insulation score was calculated for each chromosome and then normalized by the genome-wide median.

**TAD analysis**. When comparing TAD boundaries between high hyperdiploid ALL and ETV6/RUNX1-positive cases, boundaries detected in different samples within ±100 kb were called overlapped boundaries while the recurrent boundaries found only in one subgroup of leukemia samples were defined as subgroup-specific boundaries. To investigate loss of TAD boundaries in high hyperdiploid ALL, the multiinter command in BEDTools software was used to identify the overlapped boundaries and subgroup-specific boundaries. To further correlate the presence of boundaries in the different subgroups of leukemia, the subgroup-specific boundaries were manually traced. Briefly, balanced-corrected Hi-C matrices were plotted using Juicebox[36] and subgroup-specific boundaries were stratified into three categories, (i) boundaries showing sharp visual contrast between within and across TAD interaction frequencies were classified as strong boundaries, (ii) boundaries showing little visual contrast were classified as weak boundaries and (iii) boundaries that totally disappeared in one of subgroup of leukemia cases were classified as lost boundaries. To detect CTCF and RAD21 binding sites occupancy over the subgroup-specific boundaries, peak files were downloaded from the UCSC database (http://hgdownload.cse.ucsc.edu/goldenPath/hg19/encodeDCC/wgEncodeAwgTfbsUniform/) and the intersectBed command in BEDTools software was used to identify all CTCF/RAD21 peaks located within a ±50 kb window around the boundary.

**Chromosome morphology analysis**. Bone marrow or peripheral blood preparations and G-banding was performed according to standard methods from cells obtained at diagnosis and stored in fixative (methanol:acetic acid; 3:1) at −20 °C. The slides were analyzed using an Eclipse 80i microscope (Nikon, Tokyo, Japan) equipped with a progressive scan camera (JAI, Copenhagen, Denmark) and a ×100 oil immersion Plan Apo VC lens (Nikon). Metaphases were captured, edited and karyotyped with the CytoVision software (Leica Biosystems, Wetzlar, Germany). Each metaphase was scored from 1–3 according to chromosome morphology, as judged by the level of chromosome condensation, the band resolution, overall chromosome shape and clearness, and how easily the chromosome pairs could be identified.

The criteria used for scoring were: 1—poor chromosome morphology, where no substantial banding pattern could be observed, chromosomes were very condensed, chromosomes presented a fuzzy appearance, i.e., chromosome shape was poor, and homolog pairs were difficult to identify; 2—fair morphology, where band level was at 200–300, chromosomes were less constricted and presented a sharp appearance, which made it easier to karyotype; and 3—good morphology, where the band levels was at least 350–400, chromosomes were elongated and presented an ideal appearance for cytogenetic analysis (Fig. 7). The person doing the chromosome morphology investigation was blinded to the results from Hi-C analysis.

**Reporting Summary**. Further information on experimental design is available in the Nature Research Reporting Summary linked to this article.

## Data availability

The mass spectrometry proteomics data have been deposited to the ProteomeXchange[70] Consortium (http://proteomecentral.proteomexchange.org) with the dataset identifier PXD010175. RNA-seq data have been deposited to the European Genome-phenome Archive (EGA) under the accession code EGAS00001003079. The remaining data will be available for academic research on somatic variants only by contacting the authors. Publicly available data used in this study can be found as deposited in the following datasets: Oligo(dT) RNA-seq data for ALL patients, accession number EGAD00001002112. Expression data from ALL patients, accession numbers GSE13351 and GSE13425. RNA-seq dataset for AML, accession number TCGA-LAML. RNA-seq dataset for papillary renal cell carcinoma, accession number TCGA-KIRP. Hi-C datasets for GM12878 cell line and IMR90 cell line, accession number GSE63525. GM12878 CTCF ChIA-PET dataset, accession number GSM1872886. GM12878 RAD21 ChIA-PET dataset, accession number GSM1436265.

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

## Acknowledgements

This study was supported by grants from the Swedish Cancer Society (K.P., grant references CAN 2014/1258 and CAN 2016/497), the Swedish Childhood Cancer Foundation (M.Y., grant reference TJ2016-0063; M.V. grant reference TJ2014-0063; R.J., grant references TJ2016-0035 and PR2016-0019; J.L., grant reference PR2016-0059; K.P., grant reference PR2015-0012), the Swedish Research Council (R.J., grant reference 2017-01653; J.L., grant reference 2015-04622; K.P., grant reference 2016-01459), the Royal Physio-graphic Society of Lund (M.Y.), Felix Mindus Contribution to Leukemia research (M.V. and R.J.), Dr Åke Olsson Foundation for Hematological Research (R.J., grant reference 2017-00437), Governmental Funding of Clinical Research within the National Health Service (K.P., grant reference ALFSKANE-623431), Cancer Research UK (L.H., D.T.O., grant reference 20412), the Wellcome Trust (L.H., D.T.O., grant reference 202878/A/16/Z) and the European Research Council (D.T.O. grant reference 615584).

## Author contributions

M.Y., M.V., J.L. and K.P. conceived the study. M.Y. analyzed RNA-seq, proteome, WES, and Hi-C data. M.V. and R.J. performed the MS experiments. M.V. and I.S analyzed proteome and RNA-seq data. L.H.M.-C. performed metaphase chromosome experiments. A.C. provided clinical data and input. T.F. supervised RNA-seq. H.L. performed RNA-seq. D.T.O. supervised Hi-C experiments. L.O. performed SNP array analyses. N.R. and E.L.W. performed RNA-seq and WES. L.H. performed Hi-C experiments and analyzed data. J.L. and K.P analyzed data and supervised the study. M.Y., M.V., J.L. and K.P. wrote the manuscript with input from all authors.

## Additional information

**Competing interests:** The authors declare no competing interests.

