## [Peer Review File · Nature Communications]

Reviewers' comments:

Reviewer #1 (Remarks to the Author):

The authors analyzed 48 high hyperdiploid and 41 ETV6-RUNX1 positive ALL cases, using whole genome/exome sequencing, SNP array and RNA-sequencing. 18 high hyperdiploid and 9 ETV6-RUNX1 cases were also analyzed by Mass Spectrometry.

For the Mass Spec analysis: 8222 proteins were analyzed (and for those also mRNA values were available). Overall, there was a good correlation between mRNA levels and protein levels, but this was less or not true for members of the ribosome or spliceosome.

The authors observed that hyperdiploidy is associated with dosage effects of the genes (due to chromosomal copy number changes). However, they also noted that not all genes on the chromosome behave in the same way, and that not all genes of duplicated chromosomes show higher expression at mRNA or protein level (and even showed negative correlation).

Next, the authors determined the use of proteomics to distinguish high hyperdiploid from ETV6/RUNX1-positive leukemia, which was possible both by proteomics and transcriptomics. >2000 proteins were up- or down-regulated between the two types of leukemia, including some that were previously linked with one of these subtypes (FLT3, CD44, IGF2BP1, RAG1/2, etc.) The authors correctly conclude that the top differentially 183 expressed proteins obtained by Mass Spec analysis agree well with previously reported RNA expression results.

Finally, by performing Hi-C analysis, metaphase chromosome morphology and associations with gene expression / protein levels, the authors conclude that the data suggests that high hyperdiploid ALL exhibits an aberrant gene expression pattern that could be associated with loss of insulation at TAD boundaries.

Major remarks:

- Overall, the study is of high experimental level, and a large number of cases has been analyzed. The correlation (or lack of correlation) between protein data and mRNA data is of high interest to further understand ALL biology. A major limitation of the study is that it remains descriptive and no direct consequences of the results for diagnosis or treatment nor any very new biological insight is provided. It would have been very interesting to see an in-depth analysis of part of the data being worked out toward a new understanding of why there is not always correlation between RNA and protein data, for example.
- Nothing is confirmed: for example: does knock-down of CTCF in ETV6-RUNX1 samples have an effect on chromosome structure and for example on the image one gets after metaphase chromosome morphology ?
- All comparisons of expression levels are done between the 2 subtypes of leukemia studied. If protein A is low in high hyperdiploid cases compared to ETV6-RUNX1 cases, it does not necessarily mean that protein A is very low in high hyperdiploid. This is true for most comparisons done in this study. Comparisons to normal B-cell subsets would be needed to make stronger conclusions.

Minor remarks:

- There was a good correlation between mRNA levels and protein levels, but this was less or not true for members of the ribosome or spliceosome: were only protein coding RNAs considered ? the ribosome and the spliceosome contain also non-coding RNAs that function as RNA, there is no protein expected for these: were these non-coding RNAs excluded from this analysis ?

Reviewer #2 (Remarks to the Author):

In this manuscript the authors perform an integrated analysis of preB cell leukemia comparing high hyper diploid with ETV6/RUNX1 fusion subgroups. They perform mass spectrometry-based proteomic analyses on 27 of which 18 belong to the hyperdiploid category. Hi-C is performed on 6 (4 hyperdiploid) of which 3 has proteomic data and ribozero RNAseq data.

The main findings are:

1. Confirmation that RAS pathway genes are mutation in hyperdiploid BCP ALL.
2. As reported in the literature there is a modest correlation of mRNA and proteomic derived gene expression.
3. Copy number gains of whole chromosomes are associated with higher mRNA and protein expression for a subset of chromosomes.
4. Hyperdiploid and ETV6/RUNX1 leukemias have distinct expression signatures at the mRNA and protein level and the differentially expressed genes are correlated.
5. Hyperdiploidy is associated with increased translation, cytokine production and protein folding whereas the ETV6/RUNX1 leukemias have higher chromatin organization and G2/M checkpoint genes.
6. They find lower levels of expression of CTCF and some cohesion genes in hyperdiploid BCP ALL.
7. Pairs of genes in the same TAD have correlated in ETV6/RUNX1 leukemias and lose correlation in hyperdiploid BCP ALL.
8. In 2/5 hyperdiploid BCP ALL leukemia there is loss of some topologically associating domains at higher resolution.
9. The hyperdiploid ALL samples have a poorer quality metaphase chromosome morphology, which is attributed by the authors due to a uncharacterized effect of lower CTCF.

The manuscript contains a lot of high quality data and generally well written. The finding of loss of correlation in hyperdiploid BCP ALL within a TAD is novel and intriguing. However, the manuscript is generally descriptive in nature, and hampered by the small number of samples particularly in those where Hi-C was performed. Casing point is that loss of topologically associating domains at 25kb resolution only occurs in 2/4 hyperdiploid samples. Also, the authors will need to give more convincing proof that lower CTCF and some cohesion genes are the primary drivers of differences in gene expression between hyperdiploid and ETV6/RUNX1 fusion driven BCP ALL.

Other minor points are:

1. Figure 1: explain in legend what PSM mean.
2. Figure 2, it is not clear to me here the "red diagonal" stripes are as I don't see it.
3. Figure 2 . divide into a,b,c and d.
4. Figure 4b clarify which genes which if this was calculated from all 6 samples that had Hi-C performed.
5. Figure 6 needs a more detailed explanation as to the interpretation. Please clarify which samples have low directionality index and insulation scores. Are they HeH_9 and HeH_10?
6. The comment "to the best of our knowledge, only primary tumor samples from colon and rectal, prostate, and breast cancer have previously been subjected to proteogenomic analyses to this level" is not correct given the recent publication of PMID: 30146332 by Stewart et. al.
7. The methods should mention the blast content of the marrow/blood samples analyzed.
8. A large bulk of the references were missing, maybe due to a formatting error which will need to be addressed.

Javed Khan MD
Deputy Chief , Genetics Branch
Senior Investigator, Oncogenomics Section
Center for Cancer Research

National Cancer Institute, NIH
37 Convent Drive,
Building 37, Room 2016B
Bethesda, Maryland 20892

Response to Reviewers

“Transcriptional dysregulation and changes in chromatin architecture in high hyperdiploid childhood acute lymphoblastic leukemia”

Yang and Vesterlund et al.

Considered for publication in Nature Communications

We thank the Reviewers for their constructive criticism that has significantly improved the manuscript. We have addressed their comments as detailed below and as marked by red text in the revised manuscript. In addition, the manuscript has been formatted according to Nature Communications style and accession codes for deposited data have been added.

Reviewer #1.

The authors analyzed 48 high hyperdiploid and 41 ETV6-RUNX1 positive ALL cases, using whole genome/exome sequencing, SNP array and RNA-sequencing. 18 high hyperdiploid and 9 ETV6-RUNX1 cases were also analyzed by Mass Spectrometry.

For the Mass Spec analysis: 8222 proteins were analyzed (and for those also mRNA values were available). Overall, there was a good correlation between mRNA levels and protein levels, but this was less or not true for members of the ribosome or spliceosome.

The authors observed that hyperdiploidy is associated with dosage effects of the genes (due to chromosomal copy number changes). However, they also noted that not all genes on the chromosome behave in the same way, and that not all genes of duplicated chromosomes show higher expression at mRNA or protein level (and even showed negative correlation).

Next, the authors determined the use of proteomics to distinguish high hyperdiploid from ETV6/RUNX1-positive leukemia, which was possible both by proteomics and transcriptomics.

> 2000 proteins were up- or down-regulated between the two types of leukemia, including some that were previously linked with one of these subtypes (FLT3, CD44, IGF2BP1, RAG1/2, etc.)

The authors correctly conclude that the top differentially 183 expressed proteins obtained by Mass Spec analysis agree well with previously reported RNA expression results.

Finally, by performing Hi-C analysis, metaphase chromosome morphology and associations with gene expression / protein levels, the authors conclude that the data suggests that high hyperdiploid ALL exhibits an aberrant gene expression pattern that could be associated with loss of insulation at TAD boundaries.

Major remarks:

-Overall, the study is of high experimental level, and a large number of cases has been

analyzed. The correlation (or lack of correlation) between protein data and mRNA data is of high interest to further understand ALL biology. A major limitation of the study is that it remains descriptive and no direct consequences of the results for diagnosis or treatment nor any very new biological insight is provided. It would have been very interesting to see an in-depth analysis of part of the data being worked out toward a new understanding of why there is not always correlation between RNA and protein data, for example.

Response: *We share the reviewer's opinion that the lower than expected correlation is interesting and as suggested by the Reviewer, we have now added a detailed analysis of the underlying reason for this. Ultimately, we found that multiple factors contribute to potentially lowering the correlations, including both regulatory (e.g. miRNA and proteasomal degradation) and activity-related ones (e.g. subcellular localization). To include these data, the first Results section has been rewritten and a new Results section and Supplementary Figure have been added to the manuscript (Results pages 5-7, new Supplementary Fig. 2 and Methods pages 26-27).*

-Nothing is confirmed: for example: does knock-down of CTCF in ETV6-RUNX1 samples have an effect on chromosome structure and for example on the image one gets after metaphase chromosome morphology?

Response: *Respectfully, we disagree with the statement that "nothing" has been confirmed. We have gone to great lengths to carefully validate the findings in the study. For example, all expression analyses have been performed in three different in-house RNA and protein datasets, showing excellent correlation, and the top hits have been validated in publicly available datasets. Furthermore, RNA-seq, Hi-C, and cytogenetics all support a gene dysregulation associated with aberrant topologically associating domain structure. As regards knockdown of CTCF, several groups have shown that CTCF knockdown affects chromatin structure in interphase using multiple cell lines (see for example Tark-Dame et al PLoS Comput Biol 2014; Zuin et al PNAS 2014, Nora et al Cell 2017); repeating those experiments in an ETV6/RUNX1-positive cell line is unlikely to yield different results. Admittedly, the effect on metaphase chromosomes has been less well studied, but we deem knock-down experiments out-of-scope for this particular study and hope to be able to address the issue of CTCF and metaphase chromosome morphology further in a more thorough manner in future investigations.*

-All comparisons of expression levels are done between the 2 subtypes of leukemia studied. If protein A is low in high hyperdiploid cases compared to ETV6-RUNX1 cases, it does not necessarily mean that protein A is very low in high hyperdiploid. This is true for most comparisons done in this study. Comparisons to normal B-cell subsets would be needed to make stronger conclusions.

Response: *We agree that comparisons with normal B-cell subsets would add valuable information. However, when comparing leukemic cells with corresponding normal cells the list of differentially expressed genes will be dominated by the strong "general" leukemic signature largely associated with growth rate differences, making it difficult to draw any conclusions on the specific leukemic subtypes. We believe that our comparison between two*

leukemic subtypes gives a more detailed and relevant view of the processes that are up-or down-regulated in these leukemias, giving valuable data on weaknesses that may be exploited in novel therapies, such targeting of protein folding and proteolytic pathways in hyperdiploid ALL. To strengthen our conclusions, however, we have now added a comparison with RNA-sequencing data from sorted pro-B/pre-B cells for the top differentially expressed genes, confirming their high/low expression at the RNA level also in comparison with normal B-cell subsets (Results pages 9 and 11, Supplementary Table 4, Supplementary Figures 4 and 5, and Methods page 25).

Minor remarks:

There was a good correlation between mRNA levels and protein levels, but this was less or not true for members of the ribosome or spliceosome: were only protein coding RNAs considered ? the ribosome and the spliceosome contain also non-coding RNAs that function as RNA, there is no protein expected for these: were these non-coding RNAs excluded from this analysis ?

Response: *Only protein-coding RNAs were included in this analysis.*

Reviewer #2

In this manuscript the authors perform an integrated analysis of preB cell leukemia comparing high hyper diploid with ETV6/RUNX1 fusion subgroups. They perform mass spectrometry-based proteomic analyses on 27 of which 18 belong to the hyperdiploid category. Hi-C is performed on 6 (4 hyperdiploid) of which 3 has proteomic data and ribozero RNAseq data.

The main findings are:

1. Confirmation that RAS pathway genes are mutation in hyperdiploid BCP ALL.
2. As reported in the literature there is a modest correlation of mRNA and proteomic derived gene expression.
3. Copy number gains of whole chromosomes are associated with higher mRNA and protein expression for a subset of chromosomes.
4. Hyperdiploid and ETV6/RUNX1 leukemias have distinct expression signatures at the mRNA and protein level and the differentially expressed genes are correlated.
5. Hyperdiploidy is associated with increased translation, cytokine production and protein folding whereas the ETV6/RUNX1 leukemias have higher chromatin organization and G2/M checkpoint genes.
6. They find lower levels of expression of CTCF and some cohesion genes in hyperdiploid BCP ALL.
7. Pairs of genes in the same TAD are correlated in ETV6/RUNX1 leukemias and lose correlation in hyperdiploid BCP ALL.
8. In 2/5 hyperdiploid BCP ALL leukemia there is loss of some topologically associating domains at higher resolution.

9. The hyperdiploid ALL samples have a poorer quality metaphase chromosome morphology, which is attributed by the authors due to a uncharacterized effect of lower CTCF.

The manuscript contains a lot of high quality data and generally well written. The finding of loss of correlation in hyperdiploid BCP ALL within a TAD is novel and intriguing. However, the manuscript is generally descriptive in nature, and hampered by the small number of samples particularly in those where Hi-C was performed. Casing point is that loss of topologically associating domains at 25kb resolution only occurs in 2/4 hyperdiploid samples.

Response: *Unfortunately, we could not perform Hi-C on more primary leukemia samples due to lack of material and the high costs of these analyses. However, the correlation analyses done on inter- and intra-TAD gene pairs (Figure 4 and Supplementary Figure 8) strongly suggest a gene dysregulation associated with an aberrant TAD structure in hyperdiploid leukemia as a group, consistent across different datasets. In line with this, Hi-C showed loss of TAD boundary strength based on directionality index in two cases (HeH_9 and HeH_10) and reduced TAD insulation based on insulation score analysis in three cases (HeH_9, HeH_10 and HeH_48). Thus, only one hyperdiploid primary sample (HeH_42) did not show aberrant TAD structure in any of the analyses. We believe that the correlation analyses and the Hi-C data together clearly show that the majority of hyperdiploid leukemias display aberrant chromatin architecture.*

Also, the authors will need to give more convincing proof that lower CTCF and some cohesion genes are the primary drivers of differences in gene expression between hyperdiploid and ETV6/RUNX1 fusion driven BCP ALL.

Response: *We have now performed additional analyses showing that: 1.) Differentially expressed genes were enriched for CTCF and cohesin binding sites and that the fold changes of differentially expressed genes corresponded to the number of CTCF/cohesin binding sites. 2) A significantly higher proportion of differentially expressed genes are located close to CTCF/cohesin anchors (Results pages 12-13, new Supplementary Figures 6 and 7, Methods page 29). Taken together with our previous analyses of general gene expression in relation to expected TAD structure as well as our Hi-C data we believe that these data clearly show that differences in gene expression between hyperdiploid and ETV6/RUNX1-positive leukemia can partly be explained by lower levels of CTCF and cohesin in the former subgroup.*

Other minor points are:

1. Figure 1: explain in legend what PSM mean.

Response: *“PSM” in the figure has been changed to “Peptide spectrum matches”.*

2. Figure 2, it is not clear tom me here the “red diagonal” stipes are as I don’t see it.

Response: *We have increased the pixel size of the data points to make them more visible.*

3. Figure 2 . divide into a,b,c and d.

Response: *We were not entirely sure what the Reviewer meant with this comment. If it was to divide the c panel into c and d we believe that this would confuse the readers, as the lower panels should be directly compared with each other.*

4. Figure 4b clarify which genes which if this was calculated from all 6 samples that had Hi-C performed.

Response: *The analysis shown in Fig 4B included all cases in the oligo(dT) dataset, not only the samples that were included in the Hi-C analysis. This has now been clarified in the figure legend as well as in the Results section (page 13).*

5. Figure 6 needs a more detailed explanation as to the interpretation. Please clarify which samples have low directionality index and insulation scores. Are they HeH_9 and HeH_10?

Response: *We have expanded the legend to include an interpretation of the results and added which cases that we refer to (HeH_9, HeH_10 and HeH_48).*

6. The comment “to the best of our knowledge, only primary tumor samples from colon and rectal, prostate, and breast cancer have previously been subjected to proteogenomic analyses to this level” is not correct given the recent publication of PMID: 30146332 by Stewart et. al.

Response: *We have added the above reference to the revised manuscript (page 17).*

7. The methods should mention the blast content of the marrow/blood samples analyzed.

Response: *This information has been added to Supplementary Table 1.*

8. A large bulk of the references were missing, maybe due to a formatting error which will need to be addressed.

Response: *We apologize for this error, which has now been corrected.*

Reviewers' comments:

Reviewer #1 (Remarks to the Author):

The authors have addressed my questions/comments.
I have no new comments.

Reviewer #2 (Remarks to the Author):

The authors have added additional data analyses. However, although the manuscript contains a lot of high quality data and the finding of loss of correlation in hyperdiploid BCP ALL within a TAD is interesting, nevertheless, it is difficult to conclude that low CTCF and loss of topologically associating domain is a major driver in hyperdiploid leukemia given this is only seen in in 2/4 hyperdiploid samples and that the other 2 hyperdiploid samples show robust TAD boundaries. Also, the authors will need to give more convincing proof that lower CTCF and cohesion genes are relevant to transcriptional dysregulation and changes in chromatin architecture of hyperdiploid leukemias.

The discussion around the newly added Supplementary figure 6 and 7 are somewhat confusing as to what can be concluded from the data. The authors performed differential gene expression between hyperdiploid and ETV6/RUNX1 leukemias and identified the number of CTCF binding sites in the gene bodies and flanking 5kb regions. They find that these differential expressed genes have a slightly higher number of CTCF binding sites. They report that genes with higher number of CTCFs are both upregulated and downregulated. Similarly, there was an increased number of anchor genes among differentially expressed genes but only a modest increase in the fold change of differentially expressed genes associated with anchor gene, albeit the genes are still suppressed (Supplementary figure 7). It is not clear from this data that low CTCF in hyperdiploid tumors is a cause of the loss of TADs (this was shown in only 2/4 samples). It is not clear how the differential gene expression is linked with numbers of CTCF binding sites since this occurs in both directions (up and down). Finally, it is not clear how the expression of anchor genes is linked with differential gene expression of suppressed genes.

Finally, it is problematic that two high hyperdiploid cases (out of 4), HeH_42 and HeH_48, did not display clearly aberrant insulation at TAD borders or directionality index, and had the two highest CMS scores; and it is not possible to definitively state that they are not representative of high hyperdiploid ALL without performing additional experiments.

Response to Reviewers

Again, we would like to thank the Reviewers for their constructive criticism. We have addressed their comments as detailed below and as marked by red text in the revised manuscript.

Reviewer #1

The authors have addressed my questions/comments.
I have no new comments.

Response: We were pleased to read that the Reviewer is now satisfied with our revised manuscript.

Reviewer #2

The authors have added additional data analyses. However, although the manuscript contains a lot of high quality data and the finding of loss of correlation in hyperdiploid BCP ALL within a TAD is interesting, nevertheless, it is difficult to conclude that low CTCF and loss of topologically associating domain is a major driver in hyperdiploid leukemia given this is only seen in 2/4 hyperdiploid samples and that the other 2 hyperdiploid samples show robust TAD boundaries. Also, the authors will need to give more convincing proof that lower CTCF and cohesion genes are relevant to transcriptional dysregulation and changes in chromatin architecture of hyperdiploid leukemias.

Response: Regarding the TAD boundaries, two hyperdiploid samples showed reduced insulation based on both insulation score analysis and directionality index, one case showed reduced insulation based on insulation score analysis but not with directionality index and one case did not show reduced insulation based on either method. To make this clearer, we have now modified Fig. 6 (“zoom-in”).

As a group, hyperdiploid cases displayed transcriptional dysregulation in relation to TAD borders (Fig 4b). To further ensure that our analysis was correct, we have now also performed pairwise correlation analysis on randomly selected gene pairs with 1Mb window size. The result showed no difference among intra-/inter-TAD and randomly selected gene pairs in hyperdiploid samples, while a clear difference in correlation can be seen between intra-TAD gene pairs and inter-TAD/randomly selected gene pairs in ETV6/RUNX1-positive cases; see modified Fig 4b, Results and Methods. Taken together, this strengthens the notion that transcriptional dysregulation in hyperdiploid cases is related to TAD borders.

Since TAD borders are determined by CTCF and cohesin binding, we believe that it is highly likely that the transcriptional dysregulation seen is caused by the relatively low expression of CTCF and cohesin. To allow for alternative interpretations, however, we have now rephrased parts of the Abstract, Introduction, Results and Discussion, as well as modified the title of the manuscript as requested by the Editor.

The discussion around the newly added Supplementary figure 6 and 7 are somewhat confusing as to what can be concluded from the data. The authors performed differential gene

expression between hyperdiploid and ETV6/RUNX1 leukemias and identified the number of CTCF binding sites in the gene bodies and flanking 5kb regions. They find that these differentially expressed genes have a slightly higher number of CTCF binding sites. They report that genes with higher number of CTCFs are both upregulated and downregulated. Similarly, there was an increased number of anchor genes among differentially expressed genes but only a modest increase in the fold change of differentially expressed genes associated with anchor gene, albeit the genes are still suppressed (Supplementary figure 7). It is not clear from this data that low CTCF in hyperdiploid tumors is a cause of the loss of TADs (this was shown in only 2/4 samples). It is not clear how the differential gene expression is linked with numbers of CTCF binding sites since this occurs in both directions (up and down). Finally, it is not clear how the expression of anchor genes is linked with differential gene expression of suppressed genes.

Response: *We apologize for the unclear discussion around these figures. The aim of the additional analyses shown in Supplementary Fig. 6 was not to show that low levels of CTCF causes loss of TADs but that it underlies some of the gene expression differences between hyperdiploid and ETV6/RUNX1-positive leukemia. Considering the different genetic drivers in these subtypes – aneuploidy vs a translocation – we do not claim that the low levels of CTCF is the only factor causing gene expression differences but rather one of several contributors. Regarding the fact that the number of CTCF binding sites is associated with both up- and downregulated genes, this agrees well with the recent study by Nora et al. (Cell 2017;169:930), where acute depletion of CTCF leads to both up- and downregulation of genes in mouse embryonic stem cells, as well as with Aitken et al (Genome Biol 2018; 19:106), where hemizygous deletion of CTCF in a mouse model also lead to both up- and downregulation of genes. Regarding the data associated with Supplementary Fig. 7, the aim of this analysis was to show that genes close to CTCF and cohesin anchors were more commonly differentially expressed, which we infer to mean that their differential expression is associated with changes in the TAD structure caused by loss of CTCF and/or cohesin binding and not to CTCF acting as a transcription factor. To clarify this, we have made changes to the Results section on page 12.*

Finally, it is problematic that two high hyperdiploid cases (out of 4), HeH_42 and HeH_48, did not display clearly aberrant insulation at TAD borders or directionality index, and had the two highest CMS scores; and it is not possible to definitively state that they are not representative of high hyperdiploid ALL without performing additional experiments.

Response: *Regarding the Hi-C data, HeH_48 showed reduced insulation based on insulation score analysis but not with directionality index analysis, whereas HeH_42 did not show reduced insulation based on either method. Since the analysis done based on RNA-seq data in Fig 4b suggest that HeH as a group has reduced insulation at TAD borders (or a different TAD structure), we thought it noteworthy that these two cases were outliers in the CMS analysis. However, we now deleted this whole sentence from the manuscript (page 17).*